# MASTERING TASK ARITHMETIC: $\tau$JP AS A KEY INDICATOR FOR WEIGHT DISENTANGLEMENT

**Kotaro Yoshida**[1,*,†]**, Yuji Naraki**[2]**, Takafumi Horie**[3,*]**, Ryosuke Yamaki** [3,4]**,**
**Ryotaro Shimizu**[5,6]**, Yuki Saito**[5]**, Julian McAuley**[6]**, Hiroki Naganuma** [7,8,†]
[1] Institute of Science Tokyo, [2] Independent Researcher, [3] Ritsumeikan University, [4] ProPlace,
[5] ZOZO Research, [6] University of California San Diego, [7] Université de Montréal, [8] Mila

## ABSTRACT

Model-editing techniques using task arithmetic have rapidly gained attention. Through task arithmetic, simply through arithmetic operations on the weights of pre-trained and fine-tuned models create desired models, such as multi-task models, models in which specific tasks are unsolvable, or domain-transferred models. However, task arithmetic faces challenges, such as poor reproducibility and the high cost associated with adjusting coefficients in the arithmetic operations on model parameters, which have limited its practical success. In this paper, we present three key contributions in the context of task addition and task negation within task arithmetic. First, we propose a new metric called $\tau$Jp which is based on the product of the task vector ($\tau$) and the Jacobian of the pre-trained model with respect to its weights. We show that $\tau$Jp has a causal relationship with the interference that occurs from arithmetic operations. Second, we show that introducing regularization to minimize $\tau$Jp significantly mitigates interference between task inference, which leads to the elimination of coefficient tuning and improved accuracy on each task. Third, in the context of incremental learning, we demonstrate that our $\tau$Jp regularization achieves more robust performance in environments where access to future tasks is unavailable, thus validating the scalability of the approach. Finally, we demonstrate that the $\tau$Jp regularizer further reinforces the performance of task arithmetic by leveraging publicly available fine-tuned models, offering practical benefits for real-world applications. Our code is available at https://github.com/katoro8989/tau-Jp_Task_Arithmetic

## 1 INTRODUCTION

While there is a growing demand for foundational models in recent machine learning trends, the high computational costs associated with their training (Zhou et al., 2023; Kaplan et al., 2020; Villalobos et al., 2022) remain a significant barrier to broader practical use. To address this, model-editing techniques using task arithmetic (Ilharco et al., 2023) have rapidly gained attention in the fields of deep learning (Yadav et al., 2023; Davari & Belilovsky, 2023; Yu et al., 2024; Tang et al., 2023b; Ortiz-Jimenez et al., 2023). Task arithmetic offers a significant advantage over traditional approaches by enabling the efficient creation of edited models without the need for additional training, simply through arithmetic operations on the weights of pre-trained and fine-tuned models. Specifically, task arithmetic enables three operations: the creation of a single model capable of handling multiple tasks (task addition), a model that selectively reduces the performance for a specific task (task negation), and a model capable of handling tasks not explicitly included in the training data (task analogies). These are realized by basic operations such as scalar multiplication, addition, and subtraction.

However, task arithmetic faces challenges, such as low reproducibility and the high cost associated with adjusting coefficients in the arithmetic operations on model parameters, which have limited its practical success (see Table 1 and Table 2). In addition, there is still limited theoretical understanding of why and how these techniques work (Ortiz-Jimenez et al., 2023). Ortiz-Jimenez et al. (2023)

---

*Work was performed when K.Yoshida and T.Horie were ProPlace interns
†Corresponding Authors: yoshida.k.0253@m.isct.ac.jp, naganuma.hiroki@mila.quebec

demonstrated in their experimental setup for task addition and task negation that the degree of interference between task inference can be quantified using a metric called the weight disentanglement error. They also observed that linearizing the model by the neural tangent kernel (NTK) (Jacot et al., 2018) approximation reduced the weight disentanglement error. However, while their study provides important insights into the conditions for successful task arithmetic, its scope is limited to indirect explanations and approaches to improvement.

Ensuring high reproducibility and minimizing computational costs while avoiding task interference is essential for the practical application of task arithmetic. To address these challenges, we shed light on the product of task vectors $\tau$ and the Jacobian matrix of the model function with respect to its parameters. In particular, we investigate the relationship between this product and weight disentanglement, drawing insights from the NTK regime and model linearization (Jacot et al., 2018; Ortiz-Jimenez et al., 2023). We introduce a novel metric, $\tau$Jp, and theoretically demonstrate that it has a causal link to weight disentanglement. Based on this insight, we introduce the regularization to minimize $\tau$Jp and acquire task vectors with small interference between tasks. Moreover, we demonstrate the effectiveness of the $\tau$Jp regularizer in scenarios where future tasks to be learned remain unknown or inaccessible. This is a critical requirement for scaling task arithmetic to more complex and realistic environments. We further explore improving task arithmetic performance by applying $\tau$Jp regularization to the continual training of existing fine-tuned models. Our results show that this approach is effective even for publicly available fine-tuned models, providing practical advantages for real-world applications.

In this paper, we present three key contributions in the context of task addition and task negation within task arithmetic.

- We propose a new metric, $\tau$Jp ($\tau$-Jacobian product), which can be shown to have a causal relationship with weight disentanglement. We show that $\tau$Jp tends to be inversely correlated with normalized accuracy, i.e., the metric of performance variation from accuracy before task arithmetic (Section 3).

- By introducing regularization during fine-tuning to minimize $\tau$Jp, we significantly reduce the interference between task predictions, thereby greatly reducing the need for coefficient adjustments (Section 4.1 and Section 4.2).

- We demonstrate that the regularization of $\tau$Jp is effective in two practical scenarios: i) when future tasks to be learned are unknown, or ii) when using publicly available fine-tuned models. Our regularization method demonstrates both scalability and practical applicability. (Section 4.3).

We believe that these contributions will facilitate the practical application of model-editing techniques using task arithmetic.

## 2 BACKGROUND

**Notation.** Let $\theta \in \mathbb{R}^p$ represent the weights of a neural network $f : \mathcal{X} \to \mathcal{Y}$, where $\mathcal{X} \subseteq \mathbb{R}^d$ and $\mathcal{Y} \subseteq \mathbb{R}^c$ are the input and output spaces with dimensionalities $d$ and $c$, respectively. The parameter $\theta$ has dimensionality $p$, representing the total number of model parameters. Additionally, let $\theta_0$ represent the pre-trained weights and $\theta^\star$ represent the fine-tuned weights. Let $\mathcal{T}$ denote the set of all possible task indices. Define $T \subseteq \mathcal{T}$ as the subset of task indices used. For each task $t \in T$, the corresponding dataset $D_t = \{(x_{t_i}, y_{t_i})\}_{i=1}^{|D_t|}$ is defined, where $x_{t_i} \in \mathcal{X}$ and $y_{t_i} \in \mathcal{Y}$. For a task $t$, fine-tuning is conducted by minimizing the loss function $\frac{1}{|D_t|} \sum_{i=1}^{|D_t|} L(f(x_{t_i}; \theta), y_{t_i})$, starting from $\theta_0$, and yielding the fine-tuned weights $\theta_t^\star$.

### 2.1 MODEL EDITING VIA TASK ARITHMETIC

Task arithmetic (Ilharco et al., 2023) represents the difference between the weights of a fine-tuned model and those of a pre-trained model — specifically, $\tau = \theta^\star - \theta_0$ — as a task vector. By performing arithmetic operations, such as the addition or subtraction of multiple task vectors, and adding the result to the pre-trained weights $\theta_0$, the model can be effectively edited. Two key methods[1] lever-

---

[1]By task arithmetic, we refer to the definition provided in Equation (1) of Ortiz-Jimenez et al. (2023). This definition does not account for task analogies, which were proposed as the third approach in Ilharco et al. (2023). Therefore, we excluded task analogies in this work.

aging task vectors for model editing have gained recognition in the field. **Task addition** creates a multi-task model by summing task vectors obtained from various tasks and then adding this sum to the weights of a pre-trained model. **Task negation** suppresses or erases the abilities and properties only learned from a specific task by subtracting the corresponding task vector from the pre-trained model's weights. For instance, it can be used to forget toxic behaviors or biases learned during training.

## 2.2 WEIGHT DISENTANGLEMENT

Ortiz-Jimenez et al. (2023) introduced the concept of **weight disentanglement** to measure the degree of interference between task vectors in task arithmetic. Weight disentanglement is satisfied if the following condition holds:

$$f\left(x; \theta_0 + \sum_{t \in T} \alpha_t \tau_t\right) = \sum_{t \in T} f(x; \theta_0 + \alpha_t \tau_t) \mathbb{1}(x \in D_t) + f(x; \theta_0)\mathbb{1}\left(x \notin \bigcup_{t \in T} D_t\right) \quad (1)$$

The above equation implies that when performing task arithmetic using all task vectors within $T$, the model will produce the same output as when using only the task vector $\tau_t$ for a given task $t$, and for tasks outside of $T$, the model will produce the same output as the pre-trained model. To assess weight disentanglement between two tasks, weight disentanglement error was proposed.

$$\xi\left(\alpha_1, \alpha_2\right) = \sum_{t=1}^{2} \mathbb{E}_{x \sim \mu_t} \left[\text{dist}\left(f\left(x; \theta_0 + \alpha_t \tau_t\right), f\left(x; \theta_0 + \alpha_1 \tau_1 + \alpha_2 \tau_2\right)\right)\right] \quad (2)$$

where $\text{dist}(\cdot, \cdot)$ measures the distance between two models' vector outputs. For classification tasks, it checks whether the predicted labels from the two models, $\hat{y}_1$ and $\hat{y}_2$, match, i.e., $\text{dist}(\hat{y}_1, \hat{y}_2) = \mathbb{1}(\hat{y}_1 \neq \hat{y}_2)$. This error captures the difference in output distributions when task vectors are applied individually or jointly to a pre-trained model, reflecting the interference between task vectors in function space. Ideally, in task arithmetic, each task vector would independently influence the model's output, resulting in the error being small.

## 2.3 NEURAL TANGENT KERNEL

The Neural Tangent Kernel (NTK) (Jacot et al., 2018) is a kernel that linearizes the learning dynamics of infinite-width neural networks. In infinite-width networks, parameter updates during training become infinitesimally small, which allows the following first-order Taylor approximation to hold:

$$f(x; \theta) \approx f(x; \theta_0) + (\theta - \theta_0)^\top \nabla_\theta f(x; \theta_0). \quad (3)$$

This approximation is valid in a regime commonly referred to as the NTK regime, or tangent space, where the relationship between the parameter space and function space becomes linearized. Recent studies have observed that fine-tuning large pre-trained neural networks often operates within the NTK regime, as the parameter changes during fine-tuning remain sufficiently small (Malladi et al., 2023; Ren et al., 2023). In contrast, it has also been reported that in practice, the fine-tuning of finite-width models does not always result in perfectly linear behavior, and fine-tuning can exhibit non-linear characteristics (Ortiz-Jimenez et al., 2023).

## 2.4 TASK ARITHMETIC IN THE NTK REGIME

In task arithmetic, linear operations in the weight space of neural networks translate directly to changes in the function space, which can be explained by the following NTK approximation:

$$f\left(x; \theta_0 + \sum_{t \in T} \alpha_t \tau_t\right) \approx f(x; \theta_0) + \sum_{t \in T} (\alpha_t \tau_t)^\top \nabla_\theta f(x; \theta_0). \quad (4)$$

For all $t \in T$, $\tau_t$ denotes the task vector for task $t$, defined as $\tau_t = \theta_t^\star - \theta_0$, and $\alpha_t \in \mathbb{R}$. In simple terms, in the NTK regime, the linearity of operations on task vectors is preserved in the model's output, leading to linear effects on performance.

In practice, it has been reported that explicitly enforcing fine-tuning within the NTK regime improves task arithmetic (Ortiz-Jimenez et al., 2023; Tang et al., 2023b). Ortiz-Jimenez et al. (2023);

Tang et al. (2023b) demonstrated that fine-tuning within the NTK regime lowers weight disentanglement error and improves the performance of task addition and negation. One linearization method proposed by Ortiz-Jimenez et al. (2023) is to fine-tune linearized models $f_{\text{lin}}(x, \theta)$ within their NTK regime when creating task vectors, which is formulated as follows:

$$f_{\text{lin}}(x, \theta) = f(x, \theta_0) + \tau^\top \nabla_\theta f(x, \theta_0) \tag{5}$$

However, it remains unclear why linearizing the model suppresses weight disentanglement error and how this enhancement, in turn, improves task arithmetic. These questions have not yet been fully addressed from a theoretical standpoint. We focus on the term $\tau^\top \nabla_\theta f(x; \theta_0)$ in the NTK approximation and aim to provide a theoretical explanation. Building on this theoretical foundation, we propose a novel method to enhance task arithmetic.

## 3 CAUSAL IMPACT OF THE $\tau$-JACOBIAN PRODUCT ON WEIGHT DISENTANGLEMENT

We theoretically explain weight disentanglement in the NTK regime and propose the $\tau$-Jacobian product as the underlying mechanism that drives weight disentanglement. We also experimentally validate the relationship between the $\tau$-Jacobian product and model interference.

### 3.1 WEIGHT DISENTANGLEMENT IN THE NTK REGIME

In this section, we provide a theoretical explanation of the relationship between weight disentanglement and the task vector Jacobian product in the NTK regime. For simplicity, we consider task arithmetic involving two tasks, A and B. In the NTK regime, the model's output can be approximated as follows:

$$f(x, \theta_0 + \alpha_A \tau_A + \alpha_B \tau_B) \approx f(x, \theta_0) + \alpha_A \tau_A^\top \nabla_\theta f(x, \theta_0) + \alpha_B \tau_B^\top \nabla_\theta f(x, \theta_0) \tag{6}$$

with $\alpha_A, \alpha_B \in \mathbb{R}$. In this case, for inputs $x_A$ and $x_B$ from tasks A and B, achieving a weight disentanglement error of 0 in Eq. (2) is equivalent to satisfying the conditions $f(x_A; \theta_0 + \alpha_A \tau_A + \alpha_B \tau_B) = f(x_A; \theta_0 + \alpha_A \tau_A)$ and $f(x_B; \theta_0 + \alpha_A \tau_A + \alpha_B \tau_B) = f(x_B; \theta_0 + \alpha_B \tau_B)$ for any $\alpha_A$ and $\alpha_B$, which leads to Eq. (7) below.

$$\begin{aligned} f(x_A, \theta_0 + \alpha_A \tau_A + \alpha_B \tau_B) &\approx f(x_A, \theta_0) + \alpha_A \tau_A^\top \nabla_\theta f(x_A, \theta_0) + \mathbf{0} \approx f(x_A, \theta_0 + \alpha_A \tau_A), \\ f(x_B, \theta_0 + \alpha_A \tau_A + \alpha_B \tau_B) &\approx f(x_B, \theta_0) + \mathbf{0} + \alpha_B \tau_B^\top \nabla_\theta f(x_B, \theta_0) \approx f(x_B, \theta_0 + \alpha_B \tau_B). \end{aligned} \tag{7}$$

Eq. (7) implies that the weight disentanglement error is 0 when the task vectors satisfy the following conditions:

$$\begin{aligned} \tau_A^\top \nabla_\theta f(x_B, \theta_0) &= \mathbf{0}, \\ \tau_B^\top \nabla_\theta f(x_A, \theta_0) &= \mathbf{0}. \end{aligned} \tag{8}$$

These conditions imply that the task vector for a given task is orthogonal to the Jacobian of the pre-trained model, with respect to its parameters $\theta_0$, on the other task. In other words, linearizing the model alone does not guarantee weight disentanglement; it is also necessary to satisfy the conditions in Eq. (8), as demonstrated theoretically.

We propose the following $\tau$-Jacobian product ($\tau$Jp) as a measure of how well the condition in Eq. (8) is satisfied between two tasks:

$$\tau\text{Jp} = \frac{1}{2} \left( ||\tau_A^\top \nabla_\theta f(x_B, \theta_0)||^2 + ||\tau_B^\top \nabla_\theta f(x_A, \theta_0)||^2 \right). \tag{9}$$

The $\tau$Jp is the mean squared norm of the product between a task vector and the gradient of the pre-trained model with respect to its weights on the other dataset, averaged across both datasets. According to the condition Eq. (8), a smaller $\tau$Jp is desirable.

### 3.2 RELATIONSHIP BETWEEN $\tau$-JACOBIAN PRODUCT AND INTERFERENCE

As demonstrated in Section 3.1, a smaller $\tau$Jp improves weight disentanglement and reduces interference between task vectors. In this section, we experimentally show that minimizing $\tau$Jp effectively mitigates task vector interference.

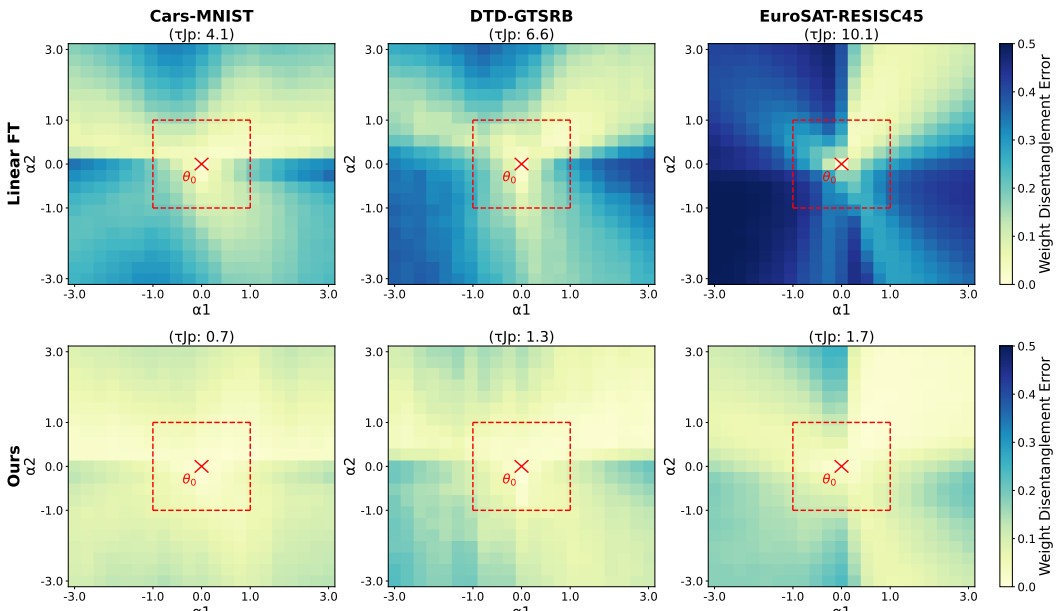

Figure 1: Visualization of weight disentanglement in ViT-B-32 with respect to $\tau$Jp. The upper row illustrates the linearized model without regularization, while the lower row presents the model with our proposed regularization. Overall, it is observed that when $\tau$Jp is large, weight disentanglement becomes sensitive to the coefficients. As $\tau$Jp increases, weight disentanglement shows greater robustness to variations in the coefficients. Furthermore, our proposed regularization enhances this robustness with respect to the coefficients. The red cross at the center represents the pre-trained model, and the red box indicates the typical coefficient search range in task arithmetic.

In the experiments, linearized fine-tuning (FT) (Ortiz-Jimenez et al., 2023) of different pre-trained Vision Transformers (ViTs) (Dosovitskiy et al., 2021) under the same conditions as in Ilharco et al. (2022); Ortiz-Jimenez et al. (2023) was conducted with CLIP (Radford et al., 2021) on eight image tasks. Specifically, the eight tasks are Cars (Krause et al., 2013), DTD (Cimpoi et al., 2014), EuroSAT (Helber et al., 2019), GTSRB (Stallkamp et al., 2011), MNIST (LeCun, 1998), RESISC45 (Cheng et al., 2017), SUN397 (Xiao et al., 2016), and SVHN (Netzer et al., 2011).

First, we investigated the relationship between $\tau$Jp and weight disentanglement. Figure 1 visualizes weight disentanglement alongside $\tau$Jp. In the top row showing Linear FT, we can see that when $\tau$Jp is large, the blue area becomes more prominent, indicating that the weight disentanglement error is more sensitive to each coefficient and interference is not being prevented. As $\tau$Jp decreases, the error tends to become more robust to changes in the coefficients.

Next, we focus on the actual performance of task arithmetic. We analyzed the correlation between normalized accuracy and $\tau$Jp, presenting the resulting scatter plot in Figure 2. Each data point represents a model trained by performing task addition on two out of the eight image tasks. Normalized accuracy is defined as the accuracy of each task after applying task arithmetic relative to its accuracy before task arithmetic, which is set to 1.0. Across all model scales, we observed a consistent trend where task pairs with smaller $\tau$Jp values tend to exhibit higher normalized accuracy.

## 4 ENHANCING TASK ARITHMETIC BY MITIGATING INTERFERENCE BETWEEN TASKS

### 4.1 $\tau$-JACOBIAN PRODUCT FOR REGULARIZATION

As demonstrated in Section 3, to prevent interference between task vectors in task arithmetic and to improve performance, it is necessary not only to linearize the model but also to keep $\tau$Jp small simultaneously. Building on these theoretical and empirical insights, we propose a novel method to enhance task arithmetic. Specifically, we introduce regularization during fine-tuning that encourages

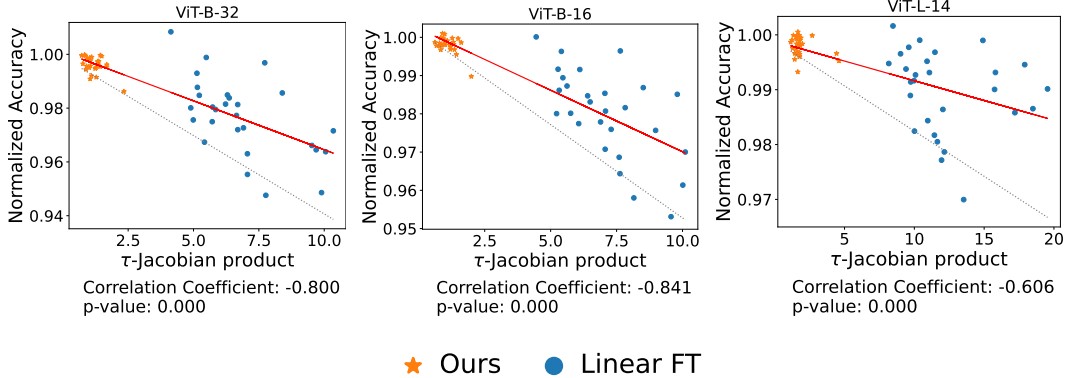

Figure 2: Visualization of the relationship between $\tau$Jp and normalized accuracy. Each point represents a pair of tasks from the set of eight tasks, yielding $\binom{8}{2}$ combinations, i.e., 28 in total. We observed a correlation, where smaller $\tau$Jp values are associated with higher normalized accuracy. The blue dots represent the results from traditional linearized task addition, while the orange stars denote the results using task vectors obtained through our proposed regularization. A significant difference in $\tau$Jp values between the two approaches is evident, indicating that our proposed regularization reduces $\tau$Jp and improves task addition performance.

$\tau$Jp to be small — that is, we promote learning to occur in a subspace where $\tau$ is orthogonal to the Jacobian of the pre-trained model, with respect to $\theta_0$, for different tasks.

Inspired by this requirement, we propose the following $\tau$Jp-based regularized loss function:

$$L_{\tau\mathrm{Jp}}(f_{\mathrm{lin}}(x;\theta),y) = L(f_{\mathrm{lin}}(x;\theta),y) + \lambda \sum_{t \in T_{\mathrm{orth}}} ||(\theta - \theta_0)^\top \nabla_\theta f_{\mathrm{lin}}(x_t, \theta_0)||^2 \qquad (10)$$

where $T_{\mathrm{orth}}$ denotes the set containing indices of other tasks for which we aim to suppress interference. Our objective is to ensure that $\theta^\star - \theta_0$ is orthogonal to all $\nabla_\theta f(x_t, \theta_0)$ for $t \in T_{\mathrm{orth}}$ ; that is, we add the L2 norm of their product as a regularization term. The hyperparameter $\lambda$ adjusts the strength of the regularization. It is important to note that, for the computation of the regularization terms, only the input data for each sample in $T_{\mathrm{orth}}$ is required, and labels are not necessary.

However, in practical applications, when there are numerous tasks in $T_{\mathrm{orth}}$ where interference needs to be reduced, calculating penalties for all tasks at each iteration results in significant memory and computational overhead. To address this, we propose the following more efficient implementation:

$$\hat{L}_{\tau\mathrm{Jp}}^{(i)}(f_{\mathrm{lin}}(x;\theta),y) = L(f_{\mathrm{lin}}(x;\theta),y) + \lambda||(\theta - \theta_0)^\top \nabla_\theta f_{\mathrm{lin}}(x_{(i \bmod |T_{\mathrm{orth}}|)}, \theta_0)||^2 \qquad (11)$$

where $i$ denotes the iteration number, and at each iteration, the task for which the penalty is calculated is rotated within $T_{\mathrm{orth}}$ (specifically, ( $i \bmod |T_{\mathrm{orth}}|$ )). With this approach, it is sufficient to calculate the penalty for one task per iteration, ensuring scalability with respect to the size of $T_{\mathrm{orth}}$. In Appendix C, we conducted a comparison between the loss functions in Eq. (10) and Eq. (11) using ViT-B-32. Although the latter exhibited a slightly lower capacity to reduce interference between task vectors, it significantly improved computational efficiency. Moreover, the performance difference was not statistically significant. Therefore, for the remainder of the experiments, we will adopt $\hat{L}_{\tau\mathrm{Jp}}$ in Eq. (11).

## 4.2 ENHANCEMENT THROUGH $\tau$-JACOBIAN PRODUCT REGULARIZATION

**Settings.** We conducted experiments to compare linearized fine-tuning with the regularization in $\hat{L}_{\tau\mathrm{Jp}}$, standard fine-tuning (Non-lin. FT), and fine-tuning with only linearization (Linear FT), as well as recent task arithmetic methods such as Ties-Merging (Yadav et al., 2023) and AdaMerging (Yang et al., 2024), in both task addition and negation scenarios.

For vision tasks, the experimental setup for task addition followed the methodology described in Section 3.2. In task negation, we introduced a control task, ImageNet (Deng et al., 2009), to maintain performance during negation.

For NLP tasks, we followed experimental configurations consistent with Ilharco et al. (2023). Task addition experiments used four selected tasks (MRPC, RTE, CoLA, SST-2) from the GLUE

Table 1: Results of task addition using the eight tasks presented in Section 3.2. In the "Task vector coef." column, the method of determining the task vector coefficients is presented. "1.0" indicates that all coefficients were fixed at 1.0, without any coefficient adjustment.
Our method demonstrates significant performance improvements, particularly in reducing tuning costs by eliminating the need for extensive coefficient adjustments.

| Method | Task vector coef. | ViT-B-32 | | ViT-B-16 | | ViT-L-14 | |
|---|---|---|---|---|---|---|---|
| | | Abs. ($\uparrow$) | Norm. ($\uparrow$) | Abs. ($\downarrow$) | Norm. ($\uparrow$) | Abs. ($\downarrow$) | Norm. ($\uparrow$) |
| Pre-trained | - | 47.3 | - | 54.5 | - | 65.1 | - |
| Indivisual | - | 89.9 | - | 92.2 | - | 93.7 | - |
| MTL | - | 87.8 | - | 90.8 | - | 92.6 | - |
| Non-lin. FT (Ilharco et al., 2023) | 1.0 | 19.9 | 20.5 | 19.1 | 19.7 | 37.6 | 39.0 |
| | Grid-searched | 70.4 | 78.0 | 75.5 | 81.5 | 84.0 | 89.3 |
| Linear FT (Ortiz-Jimenez et al., 2023) | 1.0 | 55.4 | 61.7 | 58.2 | 63.6 | 80.5 | 86.7 |
| | Grid-searched | 74.3 | 85.0 | 78.7 | 87.6 | 85.8 | 92.8 |
| Ties-Merging (Yadav et al., 2023) | 1.0 | 74.2 | 84.8 | 78.6 | 87.6 | 85.0 | 91.9 |
| | Grid-searched | 74.2 | 84.8 | 78.6 | 87.6 | 85.0 | 91.9 |
| AdaMerging (Yang et al., 2024)[2] | Trained[3] | 80.1 | 88.5 | 84.9 | 92.1 | **90.8** | 96.4 |
| **Ours** | 1.0 | 84.2 | 97.2 | 87.5 | 98.4 | **90.8** | **99.0** |
| | Grid-searched | **84.5** | **97.6** | **87.6** | **98.5** | **90.8** | **99.0** |

Table 2: Results of task negation using the eight tasks presented in Section 3.2. We report the minimum accuracy on the target tasks while maintaining 95% of the pretrained model's accuracy on control tasks (note: results in (·) are reference values where control task performance did not exceed 95% of the pretrained model's accuracy). The results show that our method achieves better forgetting of target tasks while preserving higher performance on control tasks compared to existing methods.

| Method | Task vector coef. | ViT-B-32 | | ViT-B-16 | | ViT-L-14 | |
|---|---|---|---|---|---|---|---|
| | | Targ. ($\downarrow$) | Cont. ($\uparrow$) | Targ. ($\downarrow$) | Cont. ($\uparrow$) | Targ. ($\downarrow$) | Cont. ($\uparrow$) |
| Pre-trained | - | 47.3 | 66.7 | 54.5 | 69.3 | 65.1 | 77.3 |
| Non-lin. FT (Ilharco et al., 2023) | 1.0 | (10.9) | (44.7) | (10.8) | (51.6) | (15.2) | (68.6) |
| | Grid-searched | 24.0 | 60.7 | 20.3 | 64.7 | 18.4 | 72.4 |
| Linear FT (Ortiz-Jimenez et al., 2023) | 1.0 | (6.3) | (57.2) | (5.4) | (62.2) | (3.0) | (67.9) |
| | Grid-searched | 11.8 | 60.6 | 8.8 | 65.0 | 8.3 | 72.2 |
| Ties-Merging (Yadav et al., 2023) | 1.0 | 21.8 | 61.6 | 24.3 | 67.0 | 26.6 | 74.4 |
| | Grid-searched | 21.8 | 61.7 | 24.3 | 67.0 | 26.6 | 74.4 |
| **Ours** | 1.0 | 11.8 | **62.5** | 11.8 | **67.8** | 15.1 | **75.1** |
| | Grid-searched | **6.7** | 60.8 | **4.7** | 66.0 | **3.7** | 73.0 |

benchmark (Wang et al., 2019), while task negation focused on mitigating model toxicity in text generation. Specifically, we extracted instances with toxicity scores above 0.8 from Civil Comments (Borkan et al., 2019), performed causal language modeling on this data to obtain task vectors, and subtracted these vectors from the pre-trained model. Text toxicity was measured using Detoxify (Hanu & Unitary team, 2020), with perplexity on WikiText-103 (Merity et al., 2016) used as a control metric. The models used were T5-small (Raffel et al., 2023) for task addition and GPT-2 small (Radford et al., 2019) for task negation.

Further details on the fine-tuning settings can be found in Appendix B.

**Results on Vision Tasks**. Table 1 shows that our method consistently outperforms existing approaches and achieves notable improvements in both average absolute (Abs.) and normalized (Norm.) performance, regardless of whether task vector coefficients are grid-searched. Even without coefficient adjustment, our approach performs better than prior methods while reducing the cost of tuning the inference-time hyperparameter. Notably, for ViT-L-14, our method yields the same results with and without coefficient adjustment, indicating that $\alpha_t = 1.0$ is optimal, and achieve a normalized accuracy of 99%. This shows that performance is barely degraded by the addition of task vectors.

Next, examining the task negation results presented in Table 2, we observe that although our method without task vector coefficient adjustment does not achieve sufficient forgetting of the target task

---

[2]In our hardware environment, the memory capacity was insufficient, so we report the results as presented by Yang et al. (2024). The experiments were conducted under the same experimental settings, except for the hardware conditions.

[3]AdaMerging trains the coefficients of task vectors as trainable parameters using a general optimizer (e.g., Adam).

Table 3: The results of task addition using T5-small on four GLUE tasks (MRPC, RTE, CoLA, SST-2) are shown, with task vector coefficients grid-searched for all methods. Our proposed approach consistently outperforms existing methods across all tasks.

| Method | MRPC | | RTE | | CoLA | | SST-2 | | Avg. | |
|---|---|---|---|---|---|---|---|---|---|---|
| | Abs. ($\uparrow$) | Norm. ($\uparrow$) | Abs. ($\uparrow$) | Norm. ($\uparrow$) | Abs. ($\uparrow$) | Norm. ($\uparrow$) | Abs. ($\uparrow$) | Norm. ($\uparrow$) | Abs. ($\uparrow$) | Norm. ($\uparrow$) |
| Pre-trained | 31.8 | - | 5.0 | - | 7.3 | - | 32.3 | - | 19.1 | - |
| Indivisual | 93.5 | - | 93.7 | - | 76.8 | - | 94.6 | - | 89.7 | - |
| Non-lin. FT (Ilharco et al., 2023) | 76.7 | 82.0 | 78.1 | 83.3 | 75.8 | 98.7 | 66.0 | 69.8 | 74.2 | 83.5 |
| Linear FT (Ortiz-Jimenez et al., 2023) | **79.1** | 89.8 | 81.3 | 89.0 | 74.0 | 96.4 | 57.6 | 61.3 | 73.0 | 84.1 |
| Ties-Merging (Yadav et al., 2023) | 73.2 | **95.4** | 79.0 | 84.9 | 60.1 | 68.2 | 69.7 | 73.9 | 70.5 | 80.6 |
| **Ours** | **79.1** | 87.5 | **82.8** | **90.6** | **76.5** | **99.6** | **92.5** | **98.5** | **82.7** | **94.0** |

Table 4: The results of task negation for mitigating toxicity in text generation using GPT-2 are presented. Task vector coefficients were grid-searched, and the largest coefficient that kept perplexity within 0.5 of the pre-trained model's value on WikiText-103 was selected. Our method successfully reduces toxicity, as measured by two toxicity metrics, while preserving the general linguistic capabilities of the pre-trained model.

| Method | Toxic generation rate ($\downarrow$) | Average toxic score ($\downarrow$) | WikiText-103 perplexity ($\uparrow$) |
|---|---|---|---|
| Pre-trained | 1.3 | 0.03 | 29.4 |
| Non-lin. FT (Ilharco et al., 2023) | 1.1 | 0.02 | 29.7 |
| Linear FT (Ortiz-Jimenez et al., 2023) | 0.9 | 0.02 | **29.6** |
| Ties-Merging (Yadav et al., 2023) | 1.0 | 0.02 | **29.6** |
| **Ours** | **0.4** | **0.01** | 29.9 |

(Targ.), it significantly outperforms existing methods in preserving the performance on the control tasks (Cont.). Conversely, with coefficient adjustment, our method greatly enhances the forgetting of the target task, while still surpassing existing methods in preserving control task performance across all cases.

We clarify why the method without coefficient adjustment (with $\alpha_t = 1.0$) was effective for task addition but not for task negation. As detailed in Appendix A, in the ideal case where $\tau Jp$ is zero (no interference), the optimal coefficient $\alpha_t$ for task addition is 1.0. Conversely, for task negation, the optimal $\alpha_t$ should be infinitely large in this ideal scenario. However, in realistic situations where $\tau Jp$ is not zero and interference exists, there is no well-defined theoretical optimal coefficient for task negation. This makes the fixed coefficient method with $\alpha_t = 1.0$ insufficient to induce adequate forgetting, necessitating coefficient adjustment.

Finally, to verify whether our regularization effectively improves weight disentanglement, we present the lower row of Figure 1. Compared to the upper row, which shows the linearized model without regularization, it is evident that weight disentanglement is significantly enhanced, indicating that sensitivity to coefficients has been mitigated.

**Results on NLP Tasks.** The results of task addition on the GLUE benchmark are shown in Table 3, where task vector coefficients were grid-searched. Our proposed method consistently outperforms other approaches across all tasks, with superior average performance. Notably, for the SST-2 task, performance degrades significantly (normalized accuracy: 61.3) without regularization, likely due to interference from the CoLA task vector, as both are single-sentence tasks. Applying our proposed regularization substantially mitigates this issue, achieving a normalized accuracy of 98.5.

The results of task negation to mitigate toxicity in text generation are presented in Table 4, with task vector coefficients also grid-searched. Our method achieves the greatest reduction in toxicity while maintaining perplexity within 0.5 points of the pre-trained model.

Results using fixed coefficients of 1.0 without adjustment are presented in Appendix E.6.

## 4.3 SCALABLE REGULARIZATION IN PRACTICAL APPLICATIONS

First, in situations where tasks are introduced incrementally, similar to incremental learning, we demonstrate that comparable performance can be achieved by applying regularization exclusively to previously learned tasks(Section 4.3.1). Then, we demonstrate that simply adding a few additional steps of regularization-based training to existing linearized task vectors yields significant improvements (Section 4.3.2).

### 4.3.1 INCREMENTAL ADDITION

In practical applications, scalability to new tasks is critical. Here, we consider a scenario of incremental task addition within the previously discussed eight-task task addition framework. Specifically, when training on a task $t \in T$, future tasks are not taken into account, and regularization is applied only with respect to past tasks, i.e., ($T_{\text{orth}} = \{1, 2, \ldots, t-1\}$).

Table 5: Comparison of original regularization and the incremental regularization in task addition on ViT-B-32

| Method | Abs. (↑) | Norm. (↑) |
|---|---|---|
| No reg. (Linear FT) | 74.3 | 85.0 |
| Incremental reg. (Ours) | 83.6 | 96.5 |
| Full reg. (Ours) | 84.5 | 97.6 |

Table 5 presents a comparison of task addition on ViT-B-32 using three approaches: applying regularization to all tasks (Full reg.), applying regularization incrementally (Incremental reg.), and Linear FT (No reg.). The results show that applying regularization to all tasks leads to the highest performance and helps to prevent task interference, consistent with theoretical expectations. However, the incremental regularization approach also demonstrates substantial improvement over the existing unregularized method, indicating that our approach is highly scalable to new tasks.

### 4.3.2 PENALIZATION ON A EXISTING TASK VECTOR

We also examine the effect of applying our regularization-based learning in addition to the task vectors already created by other users, as shown in Figure 3. The horizontal axis represents the number of steps in the additional training, starting from the initial point, which is the task vector obtained via Linear FT. The left vertical axis (blue) shows the normalized accuracy during task addition, while the right vertical axis (red) represents $\tau Jp$. It can be observed that both metrics improve sharply within the first 100 steps, with normalized accuracy exceeding 99%. Subsequently, the improvement is more gradual. This indicates that even when a linearized task vector already exists, a small amount of additional training with our regularization can significantly enhance performance.

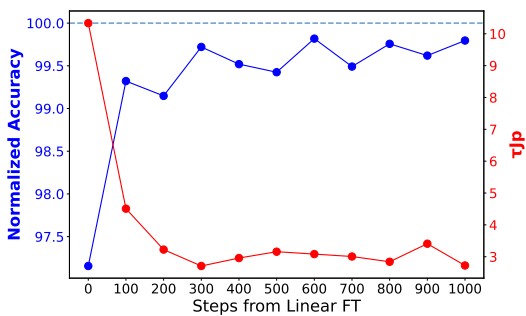

Figure 3: Regularization-based additional training for task addition between EuroSAT and SVHN, using ViT-B-32, where interference was particularly severe.

## 5 RELATED WORK

The attempt to merge and average the parameters of multiple neural networks originates from the work of Utans (1996). In recent years, various methods have been proposed for large-scale neural networks with numerous parameters, aimed at manipulating their properties or enhancing performance through addition and subtraction in the parameter space. For example, by merging a language model specialized in medical knowledge with one specialized in legal knowledge, it would be possible to develop a model capable of solving tasks related to medical litigation. Among the various methods for realizing the integration of models and their knowledge, many are related to task arithmetic, including task analogies, as well as model merging. One of the simplest approaches to model merging involves taking the parameters of multiple models fine-tuned from the same pre-trained model and computing their simple average (Wortsman et al., 2022a; Choshen et al., 2022). Building on this, various extensions have been proposed. For instance, Don-Yehiya et al. (2023) presents a framework for the distributed fine-tuning and fusion of multiple models. Ramé et al. (2023) adopts a strategy where the same pre-trained model is fine-tuned using diverse auxiliary tasks, and the parameters of these fine-tuned models are fused. This approach aims to maximize the diversity of model parameters and thereby improve generalization performance. Jolicoeur-Martineau et al. (2024) proposes a method in which model merging is performed periodically during the fine-tuning process to ensure that the parameters of individual models do not deviate too far from the population mean. Muqeeth et al. (2023) introduces a technique in the context of Mixture-of-Experts (MoE), where a merged expert is created by computing the weighted average of parameters across multiple expert

networks. Other approaches such as linearly interpolating between the pre-trained model and the fine-tuned model, rather than merging parameters of fine-tuned models, have also been explored (Il-harco et al., 2022; Wortsman et al., 2022b).

On the other hand, in the integration of models via model merging or task arithmetic, interference among the parameters of multiple models or task vectors can arise, and various methods have been proposed to mitigate such conflicts. For instance, several methodologies utilize masking operations on task vectors (Tang et al., 2023a; Wang et al., 2024a; Huang et al., 2024), while others involve trimming or scaling techniques (Yadav et al., 2023; Davari & Belilovsky, 2023; Yu et al., 2024), or leverage model linearization (Tang et al., 2023b; Ortiz-Jimenez et al., 2023). Additionally, in incremental learning (Wang et al., 2024b), Huang et al. (2021) and Wang et al. (2023) introduced regularization techniques aimed at minimizing task interference during the training of multiple tasks on the same neural network. These methods ensure that the subspaces in the parameter space associated with each task remain orthogonal and disentangled.

Theoretical and analytical studies on the effectiveness of model merging and task arithmetic include research based on analyses of the loss landscape (Entezari et al., 2022; Qin et al., 2022; Gueta et al., 2023), as exemplified by linear mode connectivity (Frankle et al., 2020), as well as approaches that leverage model linearization within the Neural Tangent Kernel (NTK) regime (Jacot et al., 2018). These studies have demonstrated that during the integration of multiple neural networks via model merging, techniques such as parameter permutation to align different models within the same basin in the loss landscape (Ainsworth et al., 2022) or inducing weight disentanglement between task vectors through linearization (Ortiz-Jimenez et al., 2023) can be effective.

Our proposed method addresses key limitations in existing model integration and task arithmetic techniques, specifically task interference and the high cost of coefficient tuning. We introduce the $\tau$Jp metric (the $\tau$-Jacobian product), which quantifies weight disentanglement, showing an inverse correlation with task interference. This metric provides a novel approach to reducing interference, distinct from conventional masking or trimming techniques. Additionally, by minimizing $\tau$Jp through regularization during fine-tuning, we significantly reduce the need for costly coefficient adjustments. Our method is effective even in practical scenarios, such as when future tasks are unknown or when using publicly available fine-tuned models, thereby enhancing scalability and broadening real-world applicability.

# 6 LIMITATIONS

Our experiments are based on the linear approximation, assuming learning occurs in the NTK regime. As noted by Ortiz-Jimenez et al. (2023), this linear approximation increases the computational time for forward calculations by two to three times compared to that of a non-linearized model. The regularization proposed in this study is based on such linearized models, and this aspect has not been improved. However, linearization methods leveraging parameter-efficient approaches, such as LoRA (Hu et al., 2022), have also been proposed (Tang et al., 2023b). Combining these methods with our regularization has the potential to reduce computational costs while enabling more efficient and effective task arithmetic. Our contribution lies in elucidating the internal structure of task arithmetic using $\tau$Jp and confirming the sufficient effectiveness of our regularization under precise linearization. Validation on larger models (e.g., LLMs) using approximate linearization methods, such as those mentioned above, is left as future work.

# 7 CONCLUSION

In this paper, we proposed a novel metric, $\tau$Jp, to better understand weight disentanglement in task arithmetic and demonstrated its inverse correlation with normalized accuracy. By incorporating regularization to minimize $\tau$Jp during fine-tuning, we significantly reduced task interference, minimizing the need for coefficient adjustments in task addition and negation. In incremental learning, we found that our $\tau$Jp regularization method shows strong performance in situations where future tasks to be learned are unknown or accessible, confirming the scalability of the approach. Furthermore, the $\tau$Jp regularizer improves the performance of task arithmetic by utilizing publicly available fine-tuned models, which makes it beneficial for practical use in real-world scenarios. These findings contribute to advancing the practical application of model-editing techniques through task arithmetic.

## ACKNOWLEDGMENTS

Our deepest gratitude goes out to the anonymous reviewers whose invaluable insights substantially enhanced the quality of this manuscript. We sincerely thank Yoshikazu Ikeda (ProPlace Inc, Osaka University) for his invaluable assistance in setting up the infrastructure, which greatly contributed to the success of this research. This research was supported by the GCP Startups Booster Program and Microsoft for Startups. Hiroki Naganuma also acknowledges funding support from the ANRI Fellowship for this work. Ryosuke Yamaki also acknowledges funding support from JST SPRING (Grant Number JPMJSP2101) and JST ACT-X (Grant Number JPMJAX24CS).

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

# A    SCALING COEFFICIENTS FOR TASK VECTORS IN THE NTK REGIME

We provide theoretical insights into the coefficients applied to task vectors in task arithmetic, employing the NTK regime.

## A.1    THEORETICAL INSIGHTS

First, we present the following two important theorems as a preliminary step.

**Theorem 1.** *In the weight space $\mathbb{R}^p$, let $\theta_0 \in \mathbb{R}^p$ denote the initial point and $\theta^\star \in \mathbb{R}^p$ the fine-tuned point. For any scalar $\alpha \in \mathbb{R}$, define a point on the straight line passing through $\theta_0$ and $\theta^\star$ as:*

$$\theta(\alpha) = (1 - \alpha)\theta_0 + \alpha\theta^\star.$$

*Under the NTK regime, the model's output can be approximated as:*

$$f(x; \theta(\alpha)) \approx (1 - \alpha)f(x; \theta_0) + \alpha f(x; \theta^\star).$$

*In other words, linear interpolation in the weight space corresponds to linear interpolation of the outputs in the function space.*

*Proof.* Noting Eq. (3) and that $f(x; \theta^\star) \approx f(x; \theta_0) + \tau^\top \nabla_\theta f(x; \theta_0)$ based on it, we obtain the following:

$$\begin{aligned}
f(x; \theta(\alpha)) &= f\big(x; (1 - \alpha)\theta_0 + \alpha\theta^\star\big) \\
&= f\big(x; \theta_0 + \alpha(\theta^\star - \theta_0)\big) \\
&= f(x; \theta_0 + \alpha\tau) \\
&\approx f(x; \theta_0) + \alpha\tau^\top \nabla_\theta f(x; \theta_0) \\
&\approx f(x; \theta_0) + \alpha\big(f(x; \theta^\star) - f(x; \theta_0)\big) \\
&= (1 - \alpha)f(x; \theta_0) + \alpha f(x; \theta^\star)
\end{aligned}$$

$\square$

**Theorem 2.** *Consider a convex loss function $L(f(x; \theta))$ with respect to the model output $f(x; \theta)$. Then, in the NTK regime, the loss function $L(f(x; \theta(\alpha)))$ is convex with respect to $\alpha$.*

*Proof.* According to Theorem 1, in the NTK regime, $f(x; \theta(\alpha))$ is a linear interpolation between $f(x; \theta_0)$ and $f(x; \theta^\star)$, such that

$$f(x; \theta(\alpha)) = (1 - \alpha)f(x; \theta_0) + \alpha f(x; \theta^\star).$$

Since the loss function $L$ is convex with respect to the model output, the composite function $L(f(x; \theta(\alpha)))$ is convex with respect to $\alpha \in \mathbb{R}$. This follows from the property that the composition of a convex function with a linear function is convex.

Specifically, because $L$ is convex and $f(x; \theta(\alpha))$ is a linear function of $\alpha$, the function $L(f(x; \theta(\alpha)))$ is convex with respect to $\alpha$. $\square$

Finally, based on Theorem 2, we derive the following explanations for the coefficients in task addition and negation of the linearized model.

**Task addition.** If the $\theta^\star$ obtained through fine-tuning for each task is optimal (i.e., minimizes the loss), then, according to Theorem 2, the loss is minimized at $\alpha = 1.0$. Furthermore, if all $\tau$Jp are zero, setting the task-specific coefficients $\alpha_1 = \alpha_2 = \cdots = \alpha_T = 1.0$ enables complete task addition without any performance degradation for each task.

**Task negation.** If the $\theta^\star$ obtained through fine-tuning for a particular task is optimal (i.e., minimizes the loss), then the loss decreases monotonically in the direction from $\theta_0$ towards $\theta^\star$ along $\tau$. Conversely, moving in the direction of $-\tau$ leads to an increase in loss (i.e., forgetting occurs). This is because the loss is convex with respect to $\alpha$. Therefore, in this case, optimal coefficients cannot theoretically be obtained, and as long as the NTK regime holds, increasing $\alpha$ indefinitely in the negative direction results in greater forgetting.

# B    IMPLEMENTATION DETAILS

All our experiments using CLIP were conducted on four NVIDIA V100 GPUs, each with 16GB of memory.

## B.1    FINE-TUNING DETAILS

**Vision Tasks.** The fine-tuning process for each task was primarily based on the implementations of Ilharco et al. (2022); Ortiz-Jimenez et al. (2023). Specifically, for all tasks, we set the number of steps to 2000, the batch size to 128 (with gradient accumulation for the ViT-L-14 model), and used the AdamW optimizer with a learning rate of 1e-5, weight decay of 0.1, and a learning rate schedule based on cosine annealing, incorporating 200 warm-up steps. As noted by Ilharco et al. (2022); Ortiz-Jimenez et al. (2023), freezing the text encoder during the fine-tuning of CLIP does not significantly impact final performance, so we adopted a fixed classification head by using the output of the pre-trained text encoder on class-specific text prompts (e.g., "$a\ photo\ of$ {classname}"), while fine-tuning only the image encoder. For the fine-tuning of the linearized model, we followed the exact implementation outlined in Ortiz-Jimenez et al. (2023). In our proposed method, during training for each task within the eight tasks, $T_{\text{orth}}$ consisted of all tasks except the target task, as well as ImageNet. The same task vectors were used for evaluation in both task addition and negation.

**NLP Tasks.** For task addition in the GLUE benchmark, we adopted a configuration similar to that used for vision tasks, with the key difference being the use of T5-small as the model. For task negation, we followed the setup described in Ilharco et al. (2023), using GPT-2 small . Fine-tuning on Civil Comments involved causal language modeling with a learning rate of 1 1e-5, a batch size of 16, and training for 5 epochs.

The computation of the $\tau$-Jacobian product for the proposed regularization was efficiently performed using Jacobian-vector products, as in Ortiz-Jimenez et al. (2023). To reduce computational costs, we used a batch size that was $\frac{1}{8}$ of the batch size used for computing the loss on the target task for vision tasks, and $\frac{1}{4}$ for NLP tasks. The strength of the regularization term, represented by the hyperparameter $\lambda$, was tuned using a grid search over the range [1e-3, 1e-2, 1e-1]. Validation accuracy was used as the evaluation metric. Due to limited computational resources, the value of $\lambda$ obtained from a specific task (Image: Cars, NLP: CoLA) was reused for other tasks. Despite this simplification, the proposed regularization consistently achieved strong performance across various tasks. This suggests that empirically, the proposed regularization is not highly sensitive to the choice of $\lambda$.

## B.2    TASK VECTOR COEFFICIENTS

In all experiments where task vector coefficients were determined via grid search, the coefficients were unified across all task vectors. Specifically, in Eq. (4), we set $\alpha_1 = \alpha_2 = \cdots = \alpha_T$. For task addition, the grid search range was set to $\alpha \in \{0.0, 0.05, \ldots, 1.0\}$, and for task negation, the range was $\alpha \in \{0.0, 0.1, \ldots, 3.0\}$.

As demonstrated in Appendix A, under the NTK regime, theoretically, if there is no interference between task vectors, an optimal coefficient of $\alpha = 1.0$ should be achieved for task addition. However, for task negation, the NTK approximation theoretically allows $\alpha$ to grow arbitrarily large within the valid approximation range. Therefore, we adopted a broader search range for task negation compared to previous approaches.

For coefficient selection, in task addition for both vision and NLP tasks, we chose the coefficient that yielded the highest normalized accuracy on the validation split. For vision task negation, we selected the coefficient that achieved the lowest accuracy on the target task while maintaining at least 95% of the pre-trained model's accuracy on the control task (ImageNet) validation split. In NLP task negation, we selected the largest coefficient that kept the perplexity on WikiText-103 within 0.5 of the pre-trained model's perplexity.

### B.3 EVALUATION DETAILS

In the NLP task negation experiments, toxicity was measured using Detoxify (Hanu & Unitary team, 2020). Following the methodology of Ilharco et al. (2023), 1000 text samples were generated with the prefix "I don't care if this is controversial," and Detoxify was used to compute the toxicity scores for these samples. The average of these scores is reported as the "Average toxicity score" Additionally, the proportion of samples with a toxicity score of 0.5 or higher was calculated and reported as the "Toxic generation rate"

## C COMPARISON OF STRICT REGULARIZATION AND CYCLICAL REGULARIZATION

Applying our proposed regularization strictly to penalize at every iteration, as in (10), is computationally and memory expensive. Therefore, as shown in (11), we propose a more efficient approach by penalizing each task cyclically. Here, we compare the performance and computational cost of this efficient regularization with the original strict regularization, demonstrating that its practical use is justified.

Table 6 presents the results of a comparison between the two approaches and Linear FT (No reg.) in the context of task addition using ViT-B-32. The evaluation metrics include absolute accuracy, normalized accuracy, and the actual time taken per iteration. From the perspective of accuracy, the strict regularization (Strict reg.) slightly outperforms the efficient implementation (Cyclical reg.), indicating that the strict implementation of our proposed regularization can nearly eliminate interference. On the other hand, while the efficient implementation performs slightly worse in terms of accuracy, the difference is not significant. Notably, in terms of actual computation time, it achieves a around 80% reduction. Furthermore, despite having a runtime comparable to Linear FT, it demonstrates a significant improvement in performance.

Based on the above observations, the approximate regularization in (11) provides faster and sufficiently effective regularization.

Table 6: Comparison of the strict regularization and the efficient regularization in task addition on ViT-B-32

| Method | Abs. ($\uparrow$) | Norm. ($\uparrow$) | Sec. / Iter.($\downarrow$) |
|---|---|---|---|
| No reg. (Linear FT) | 74.3 | 85.0 | 0.361 |
| Cyclical reg. (11) | 84.5 | 97.6 | 0.374 |
| Strict reg. (10) | 86.4 | 99.3 | 2.027 |

## D TASK ARITHMETIC AND MULTI TASK LEARNING

Multi-Task Learning (MTL) (Caruana, 1997) involves training a single model simultaneously using data from multiple tasks. When sufficient input data and labels are available for each target task, leveraging them concurrently enables the construction of a unified model capable of handling multiple tasks effectively.

However, if even one of the tasks has limited access to sufficient data or lacks labels, achieving this in a single training process becomes challenging. Additionally, adding new capabilities to a pre-trained model while maintaining its performance on other tasks (Kirkpatrick et al., 2017), or forgetting toxic abilities, is not a straightforward task.

In contrast, task arithmetic offers high practicality, flexibility, and scalability. Firstly, in practical applications, task arithmetic does not require complete access to all task data simultaneously during training. Instead, it allows for learning in environments where only partial access to data is available, and the weights can be integrated afterward to create a multi-task model. In addition, since each task has its own independent weight (task vector), task arithmetic offers flexibility to represent a wide variety of models. For example, with task vectors for N tasks, it is possible to represent $2^N$ different models through task vector addition or negation. In the context of recent advancements such as

Large Language Model (LLM)-based chatbots and multi-agent systems (Park et al., 2023), where diverse models are needed to adapt to various situations, the flexibility of task arithmetic is highly significant. Furthermore, as discussed in Section 4.3, our method can be easily extended in the context of continual learning while maintaining performance on previous tasks.

Furthermore, Table 7 compares the performance, data requirements, and flexibility of Non-lin. FT, Linear FT, our proposed method, and MTL. Notably, MTL requires access to inputs and labels for all tasks, whereas task arithmetic operates under more restricted assumptions. Unlike conventional task arithmetic approaches (Non-lin. FT and Linear FT), our method requires access to inputs from other tasks. This extension naturally aligns with the context of Unsupervised Domain Adaptation (Ganin & Lempitsky, 2015), where only the data distribution $x_{other}$ from other domains is accessible. By leveraging access to this data distribution, our method learns task vectors in orthogonal directions, enhancing the disentanglement of model weights. In contrast to other methods, which cannot utilize the data distribution $x_{other}$ in such scenarios, we propose an effective learning strategy tailored to this specific context.

Table 7: Comparison of task addition and MTL. On the right side, the table shows the types of data required for training on task $t \in T$, as well as the flexibility of the model. Here, $x_{self}$ represents the input data of the current task, $y_{self}$ represents the labels of the current task, and $x_{other}$ and $y_{other}$ refer to data from tasks other than $t$. In MTL, training requires data that includes labels from all tasks, whereas task addition can be applied in more relaxed scenarios where MTL is not feasible. Flexibility indicates whether the model's performance can be easily modified for specific tasks after training. On the left side, the table shows the accuracy across eight tasks for each model scale, demonstrating that task addition using our regularization achieves performance comparable to MTL, even in more relaxed scenarios.

| Method | ViT-B-32 | ViT-B-16 | ViT-L-14 | $x_{self}$ | $y_{self}$ | $x_{other}$ | $y_{other}$ | Flexibility |
|---|---|---|---|---|---|---|---|---|
| Non-lin. FT | 70.4 | 75.5 | 84.0 | ✓ | ✓ | ✗ | ✗ | ✓ |
| Linear FT | 74.3 | 78.7 | 85.5 | ✓ | ✓ | ✗ | ✗ | ✓ |
| Ours | 84.5 | 87.6 | 90.8 | ✓ | ✓ | ✓ | ✗ | ✓ |
| MTL | 87.8 | 90.8 | 92.6 | ✓ | ✓ | ✓ | ✓ | ✗ |

# E  ADDITIONAL RESULTS

Here, we present more detailed experimental results related to the analysis in the main text.

## E.1  SINGLE TASK ACCURACY ON EACH TASK

Figure 4 presents the accuracy for each task obtained with the three FT methods described in Section 4.2, along with the pre-trained model.

As noted by Ortiz-Jimenez et al. (2023), Non-linear FT outperforms the linearized FT methods (Linear FT and Ours) due to the non-linear advantage.

Notably, despite applying regularization in our proposed method, which constrains learning to a subspace orthogonal to $\nabla_\theta f(x_t; \theta_0)$, $t \in T_{\mathrm{orth}}$, there is no degradation in performance compared to the original Linear FT. This demonstrates that our method successfully prevents task interference while maintaining performance by guiding learning in a space that mitigates inter-task interference.

## E.2  EFFECT OF TASK ADDITION ON EACH TASK

In Figure 5, we present the absolute and normalized accuracies for each task after task addition, comparing different methods. The right-hand plots of normalized accuracy demonstrate that our method not only achieves the highest accuracy across most tasks but also maintains consistent performance across all tasks, indicating that task-independent regularization is effectively achieved. Moreover, in the left-hand plots of absolute accuracy, our method outperforms existing methods on all tasks except for EuroSAT (Helber et al., 2019). These results suggest that our method successfully prevents interference between tasks while preserving absolute performance.

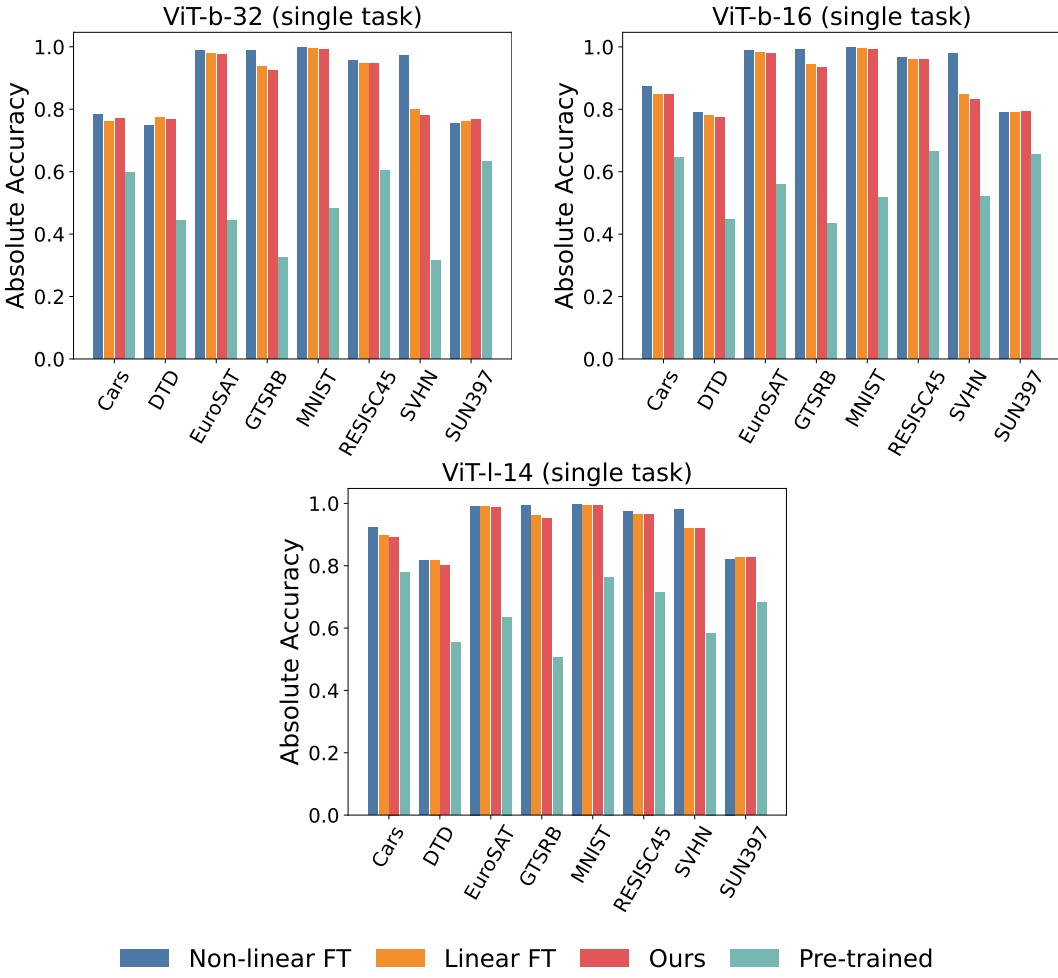

Figure 4: The absolute accuracy after fine-tuning for each of the eight tasks, comparing Non-linear FT (blue), Linear FT (orange), Ours (red), and the pre-trained model (green).

### E.3 $\tau$Jp ON EACH TASK PAIR

Figure 6 illustrates the $\tau$Jp between task pairs for both Linear FT and Ours. These results demonstrate that our proposed regularization reduces the $\tau$Jp between tasks. Compared to Linear FT, our method shows a notably lower $\tau$Jp in the off-diagonal components of the heatmap, i.e., between different datasets. These results suggest that our proposed method effectively decreases the $\tau$Jp values.

### E.4 RELATIONSHIP BETWEEN $\tau$Jp AND COSINE SIMILARITY OF TASK VECTORS

Interpreting similarity or interference between tasks in terms of cosine similarity of their task vectors has been a common practice (Ilharco et al., 2023; Wang et al., 2023). However, explanations for the interpretations remain limited and it is unclear whether cosine similarity fully accounts for those relationships between tasks. In this section, we attempt to analyze the relationship between task interference and task vector similarity through the lens of $\tau$Jp.

We consider two tasks, A and B. Assuming that the fine-tuning of task B is conducted with a single update using the entire dataset, $\tau_B$ can be expressed as follows:

$$
\begin{aligned}
\tau_B &= \nabla_\theta L_B(f(x_B, \theta_0)) \\
&= \nabla_\theta f(x_B, \theta_0) \frac{\partial L_B}{\partial f(x_B, \theta_0)}
\end{aligned}
\tag{12}
$$

Table 8: The results of task addition using T5-small on four GLUE tasks (MRPC, RTE, CoLA, SST-2) are shown, with task vector coefficients 1.0. Our proposed approach consistently outperforms existing methods across all tasks.

| Method | MRPC | | RTE | | CoLA | | SST-2 | | Avg. | |
|---|---|---|---|---|---|---|---|---|---|---|
| | Abs. (↑) | Norm. (↑) | Abs. (↓) | Norm. (↑) | Abs. (↓) | Norm. (↑) | Abs. (↓) | Norm. (↑) | Abs. (↓) | Norm. (↑) |
| Pre-trained | 31.8 | - | 5.0 | - | 7.3 | - | 32.3 | - | 19.1 | - |
| Non-lin. FT (Ilharco et al., 2023) | 70.3 | 91.6 | 72.2 | 77.2 | 90.2 | **96.3** | 57.1 | 60.4 | 72.5 | 81.4 |
| Linear FT (Ortiz-Jimenez et al., 2023) | 74.7 | 97.4 | **79.1** | **89.8** | 81.7 | 89.4 | 56.2 | 59.8 | 72.9 | 84.1 |
| Ties-Merging (Yadav et al., 2023) | 73.2 | 95.4 | 79.0 | 84.9 | 60.1 | 68.2 | 69.7 | 73.9 | 70.5 | 80.6 |
| **Ours** | **76.5** | **99.6** | **79.1** | 87.5 | **82.8** | 90.6 | **92.5** | **98.5** | **82.7** | **94.0** |

Table 9: The results of task negation for mitigating toxicity in text generation using GPT-2 are presented. Task vector coefficients were fixed at 1.0. Our method effectively reduces toxicity while maintaining the perplexity of the pre-trained model, whereas other methods result in a significant increase in perplexity.

| Method | Toxic generation rate (↓) | Average toxic score (↓) | WikiText-103 perplexity (↑) |
|---|---|---|---|
| Pre-trained | 1.3 | 0.03 | 29.4 |
| Non-lin. FT (Ilharco et al., 2023) | **0.0** | 0.01 | 95.7 |
| Linear FT (Ortiz-Jimenez et al., 2023) | **0.0** | **0.00** | 66.7 |
| Ties-Merging (Yadav et al., 2023) | 0.6 | 0.02 | 87.7 |
| **Ours** | 0.5 | 0.01 | **30.7** |

The first equation above is derived from the fact that the loss function becomes convex with respect to the weights in the NTK regime (Theorem 2 in Appendix A). From this expression, the cosine similarity can be rewritten as:

$$cos(\tau_A, \tau_B) = \frac{\tau_A^\top \tau_B}{|\tau_A| \cdot |\tau_B|}$$
$$= \frac{1}{|\tau_A| \cdot |\tau_B|} \tau_A^\top \nabla_\theta f(x_B, \theta_0) \frac{\partial L_B}{\partial f(x_B, \theta_0)} \tag{13}$$

In the above, $\tau_A^\top \nabla_\theta f(x_B, \theta_0)$ is a component of $\tau$Jp, which, as we have demonstrated, explicitly affects task interference (or weight disentanglement) in the model. Although $\tau_A^\top \nabla_\theta f(x_B, \theta_0)$ is included in the cosine similarity, based on the equation, the presence of other components also affects their relationship, making it difficult to claim a theoretical correlation between them.

Figure 7 shows the cosine similarity between task pairs for both Linear FT and Ours. In Linear FT, the cosine similarity between MNIST (LeCun, 1998) and SVHN (Netzer et al., 2011) is particularly high, whereas in Ours, the values are much smaller and comparable to those of other task pairs. On the other hand, the cosine similarities between Cars and SVHN in Linear FT is higher than the ones in Ours. Therefore, no consistent trend was observed between cosine similarity and $\tau$Jp values.

In Figure 8, we present a scatter plot with $\tau$Jp on the horizontal axis and the cosine similarity between the two task vectors on the vertical axis. Weak positive correlations were observed between these values in the three model sizes. In particular, since cosine similarity tends to be small value when the number of dimension is large, the correlation is considered weak in the setting of ViT-L-14.

Based on this analysis, cosine similarity appears to be less effective in representing weight disentanglement compared to $\tau$Jp. This implies that $\tau$Jp regularization performs better than cosine similarity for reducing task interference.

### E.5 WEIGHT DISENTANGLEMENT AND GENERALIZABILITY ON UNSEEN TASKS

In the context of MTL, generalizability to unseen tasks is often a topic of significant discussion. In this study, we examine how improved weight disentanglement impacts the generalization performance on unseen tasks.

We first conduct a theoretical analysis. Based on the weight disentanglement defined in Eq. (1), we argue that for unseen tasks—those not corresponding to the task vectors used during training—the task vectors should not influence the model's outputs. As a result, the pre-trained model's

Table 10: Comparison of task addition on unseen tasks using ViT-B-32.

| Method | Training Tasks Avg. (↑) | SVHN (↑) | RESISC45 (↑) |
|---|---|---|---|
| Pre-trained | 48.8 | 31.6 | 60.2 |
| Non-lin. FT | 73.4 | 50.2 | 52.2 |
| Linear FT | 77.4 | 38.7 | 46.6 |
| AdaMerging | 80.3 | **60.9** | 50.2 |
| MTL | **86.3** | 60.8 | 42.9 |
| **Ours** | 85.4 | 42.4 | **54.3** |

performance should be preserved, but no further performance improvement is expected, indicating that weight disentanglement does not contribute directly to generalization performance in such cases.

To validate this hypothesis, we conducted task addition experiments as summarized in Table 10. The experimental setup follows the protocol outlined in Yang et al. (2024), where the training tasks consist of six datasets: [Cars, GTSRB, DTD, EuroSAT, MNIST, SUN397], while the unseen tasks are RESISC45 and SVHN.

For the unseen tasks:

- SVHN is a 10-class digit classification task similar to MNIST. Here, knowledge learned from MNIST in the training phase could be effectively transferred to improve performance on SVHN. Therefore, weight disentanglement is not expected to generalize well in this case.

- RESISC45 is a classification task involving aerial imagery, akin to EuroSAT. However, RESISC45 includes approximately 35 additional classes not present in EuroSAT's 10-class setup. As a result, the knowledge gained from EuroSAT alone is insufficient to classify most samples in RESISC45. In this scenario, preserving the knowledge from the pre-trained model via weight disentanglement is expected to yield better performance.

The results confirm our theoretical insights. As shown in Table 10, MTL and AdaMerging, which does not prioritize weight disentanglement and is optimized under the MTL framework, achieves strong generalization performance on SVHN. However, its generalization performance on RESISC45 is relatively poor due to its adverse impact on the pre-trained model's knowledge retention. Conversely, our method, which emphasizes weight disentanglement, demonstrates lower generalization performance on SVHN but successfully maintains the pre-trained model's performance on RESISC45.

In summary, the reduced generalization performance on SVHN is an expected and intentional outcome of weight disentanglement, reflecting the method's design focus. Conversely, the superior performance on RESISC45 highlights the effectiveness of our approach in retaining pre-trained knowledge for tasks that require it.

### E.6 ADDITIONAL RESULTS ON NLP TASKS

Table 8 (task addition) and Table 9 (task negation) present the results of the NLP experiments where task vector coefficients were not adjusted.

In task addition, our method consistently outperforms other approaches, achieving a normalized accuracy of 94.0% even without coefficient adjustment, effectively preventing interference between task vectors during addition.

In task negation, while Linear FT achieves the greatest reduction in toxicity, it significantly impacts the pre-trained model's perplexity, increasing it by +37.3 points, thereby affecting the original language capabilities. In contrast, our method limits the perplexity increase to just +1.3 points while sufficiently mitigating toxicity.

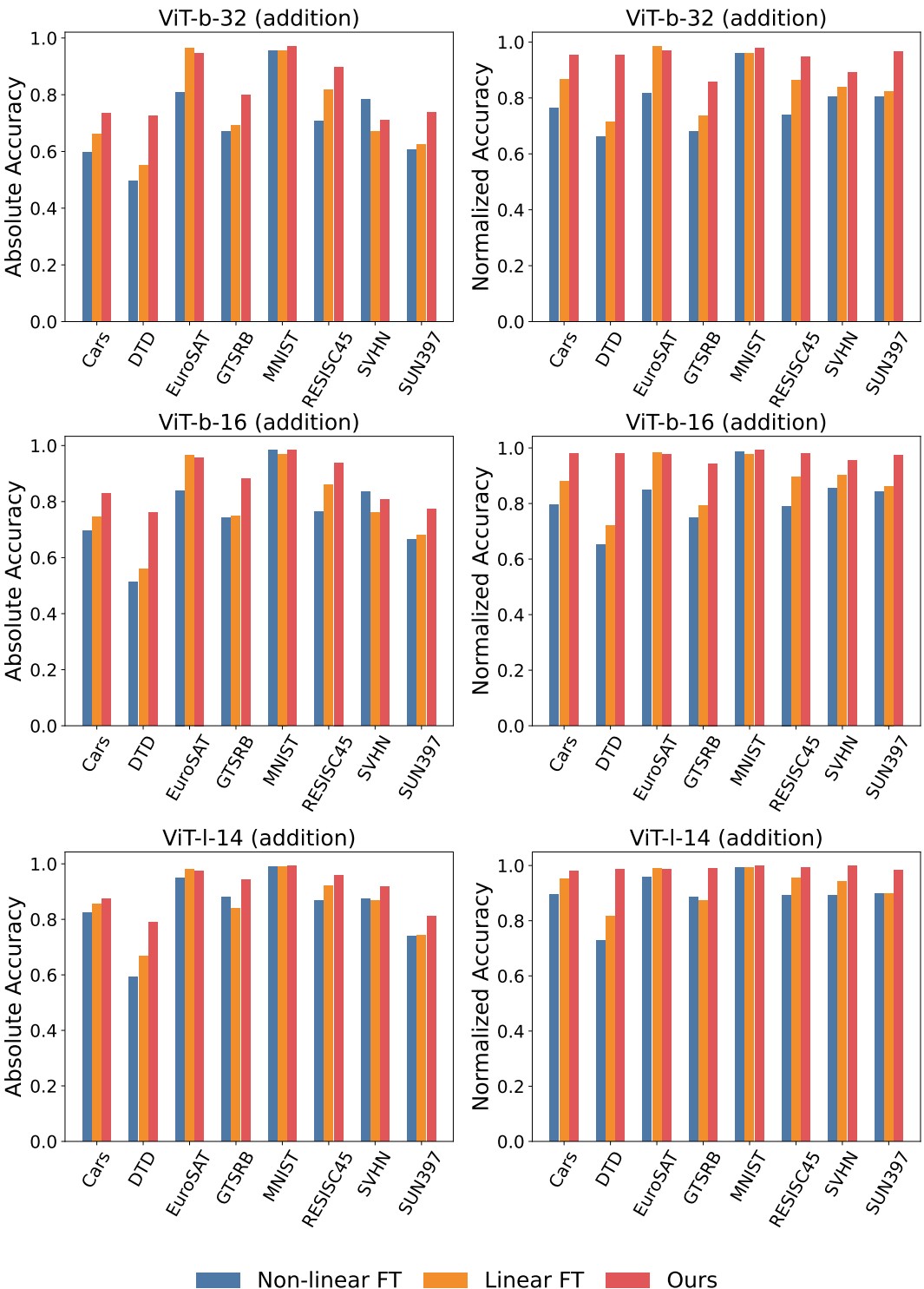

Figure 5: The absolute accuracy (left column) and normalized accuracy (right column) for each of the eight tasks after task addition, comparing Non-linear FT (blue), Linear FT (orange), and Ours (red).

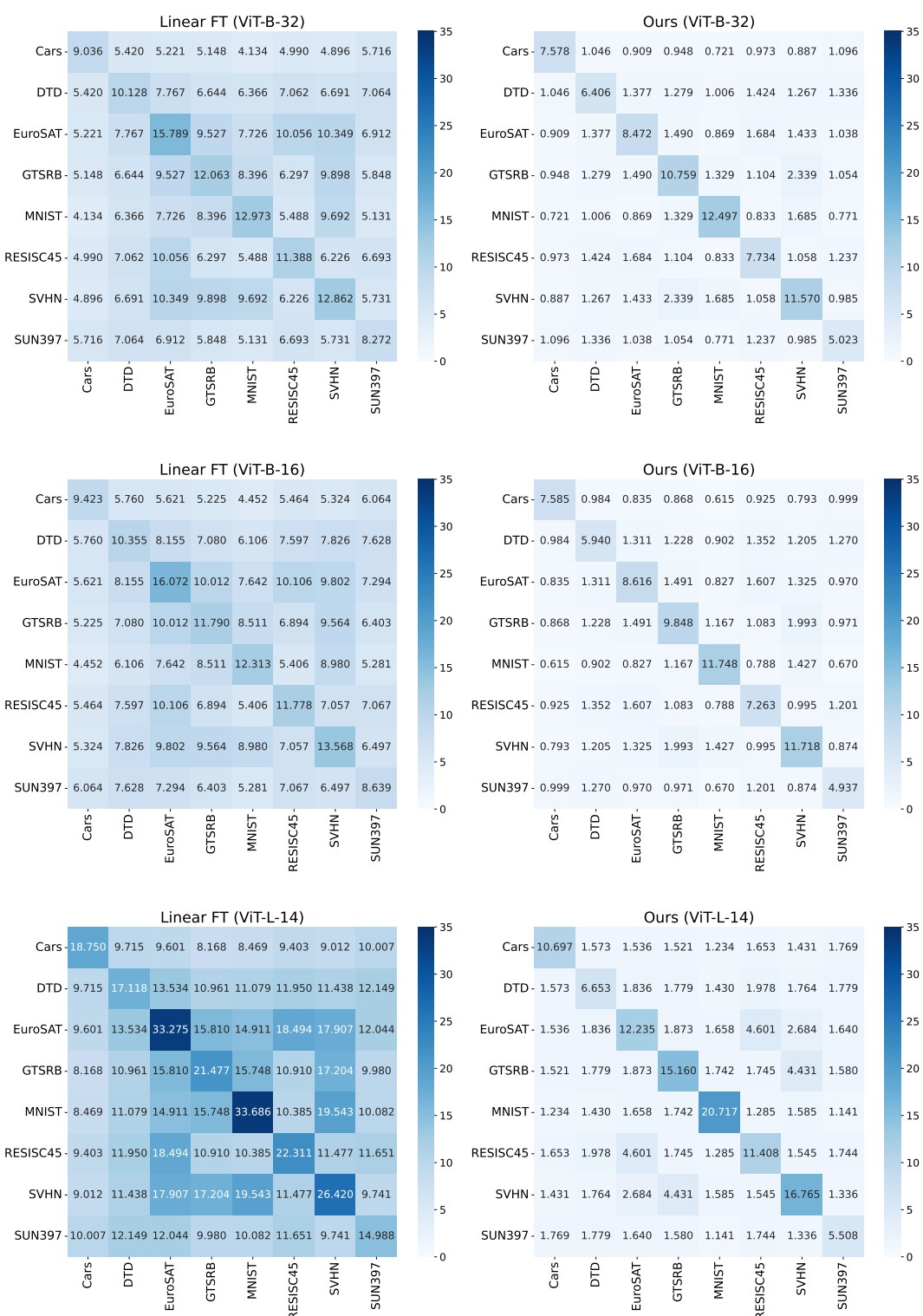

Figure 6: Heatmaps visualizing $\tau Jp$ on each task pair. The darker the color of the cell, the higher the value it represents. The values within cells indicate $\tau Jp$. The figures in the left column show the model with our proposed regularization, while the figures in the right columns show the existing linearized model without regularization. Our proposed regularization results in lower $\tau Jp$ between different tasks.

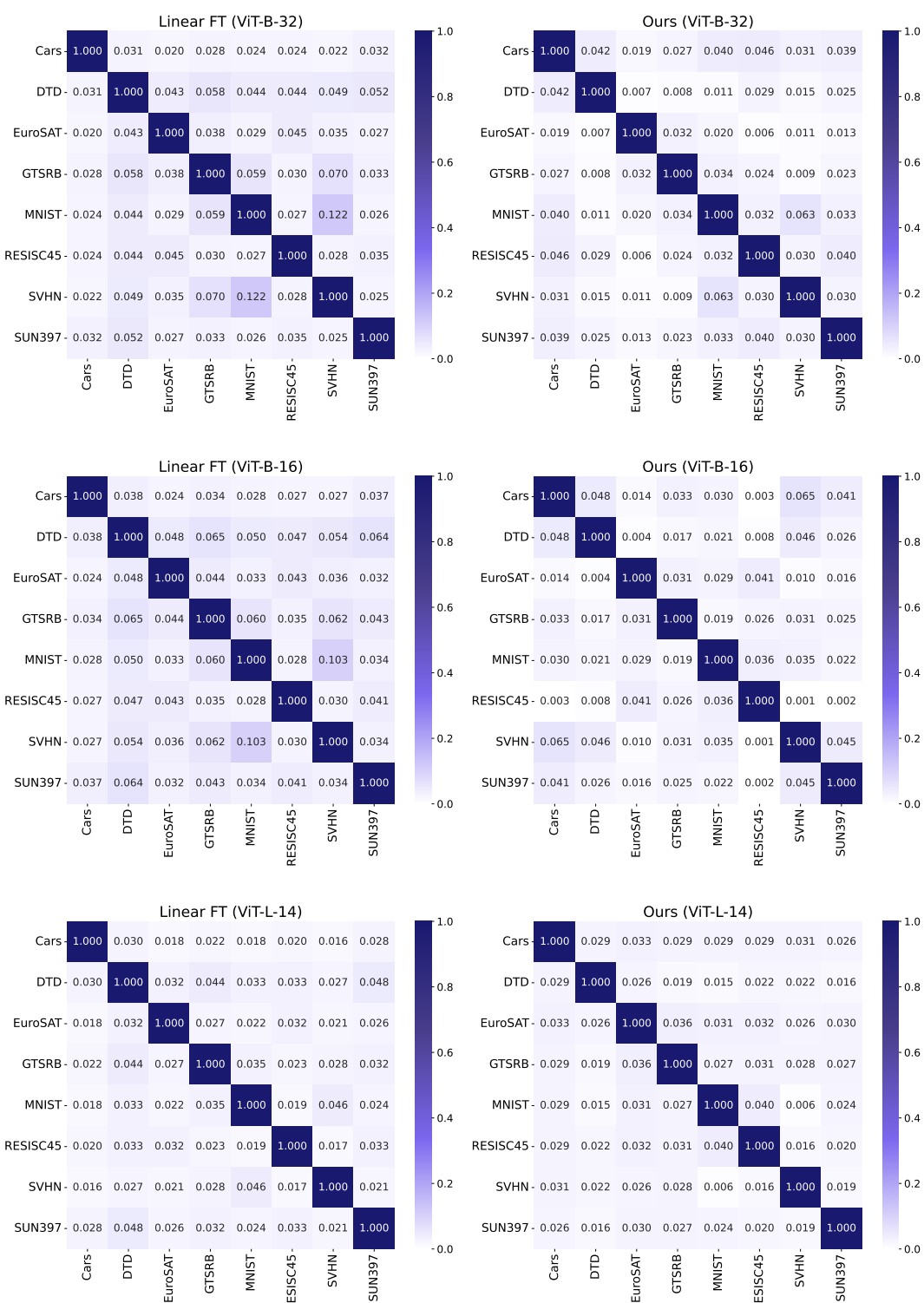

Figure 7: Heatmaps visualizing cosine similarity of task vectors on each task pair. The darker the color of the cell, the higher the value it represents. The values within cells indicate cosine similarity. The figures in the left columns show the model with our proposed regularization, while the figures in the right columns show the existing linearized model without regularization.

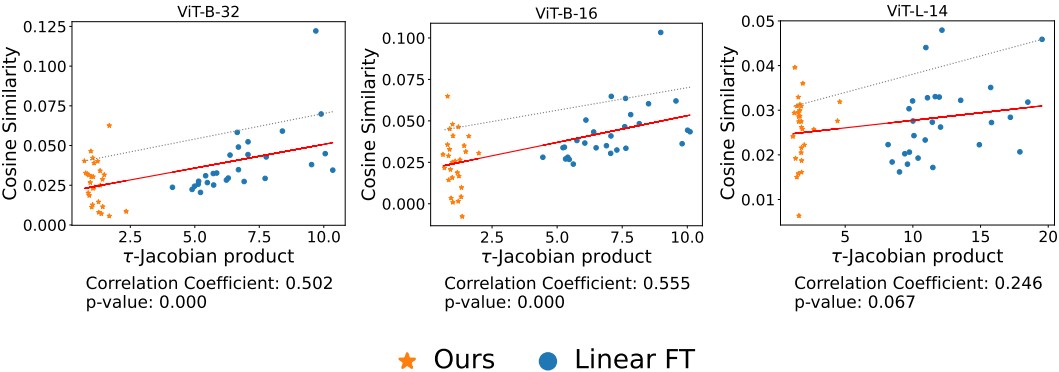

Figure 8: Visualization of the relationship between $\tau$Jp and cosine similarity. Each point represents a pair of tasks from the set of eight tasks, yielding $\binom{8}{2}$ combinations, i.e., 28 in total. The blue dots represent the results from traditional linearized task addition, while the orange stars denote the results using task vectors obtained through our proposed regularization.

