# OpenReview forum: "Mastering Task Arithmetic: $\tau$Jp as a Key Indicator for Weight Disentanglement"
_ICLR.cc/2025/Conference — ICLR 2025 Poster_

### Official Review · Reviewer_n6ZH · 2024-10-18

**Soundness:** 2
**Presentation:** 3
**Contribution:** 2
**Rating:** 5
**Confidence:** 2

**Summary:**

The paper presents a novel approach to task arithmetic in neural networks, which leverages a novel metric that quantifies the relationship between task vectors and the Jacobian of pre-trained models. The authors claim that by minimizing this metric through regularization, they can significantly reduce interference between task predictions and enhance the accuracy of task arithmetic operations. The experimental results demonstrate substantial improvements in performance for both task addition and task negation.

**Strengths:**

+ This paper is well-written and easy to follow.
+ The experiments are extensive, and the results sound good.
+ The design of metric τJp is reasonable and interesting.

**Weaknesses:**

- The novelty of this paper may be limited. My consideration is that this paper seems to fundamentally align with the approach proposed by Ortiz-Jimenez et al. (2023) [1], which also emphasizes fine-tuning models in the tangent space. Although using the specific regularization term, this paper does not sufficiently differentiate itself from this existing work.
- While the empirical results are compelling, the paper lacks a thorough theoretical explanation for why the proposed regularization leads to better performance compared to other methods, such as those discussed in Ortiz-Jimenez et al. (2023). I am confused about why a simple and soft regularization results in such improvement compared to [1]. A deeper theoretical analysis could strengthen the paper's contributions.
- The authors briefly mention tuning the regularization strength but do not provide sufficient details on how this hyperparameter was selected. The sensitive analysis of this hyperparameter is also necessary for the paper.

[1] Guillermo Ortiz-Jimenez, Alessandro Favero, and Pascal Frossard. Task arithmetic in the tangent space: Improved editing of pre-trained models. Advances in Neural Information Processing Systems, 2023.

**Questions:**

Could the proposed regularization affect the model's plasticity? Specifically, how might the addition of this regularization impact the fine-tuning performance, potentially influenced by the strength of the regularization?

---

> ### Author Response · Authors · 2024-11-22
> **Response to reviewer n6ZH**
>
> ### Regarding Weakness 1
>
> >The novelty of this paper may be limited. My consideration is that this paper seems to fundamentally align with the approach proposed by Ortiz-Jimenez et al. (2023) [1], which also emphasizes fine-tuning models in the tangent space. Although using the specific regularization term, this paper does not sufficiently differentiate itself from this existing work.
>
> Thank you for your valuable insights regarding the novelty of our contributions.
>
> In [1], it was experimentally shown that model linearization improves weight disentanglement. Furthermore, in their discussion on kernel methods, they theoretically demonstrated that the localization of kernel eigenfunctions to specific tasks leads to weight disentanglement. However, their work does not provide a methodological explanation for achieving task localization of the kernel, leaving their findings as a result-oriented discussion without a clear procedural basis.
>
> In contrast, our work not only incorporates model linearization but also theoretically establishes that reducing the $\tau_{\text{Jp}}$ term corresponding to other tasks in the NTK approximation is necessary for achieving weight disentanglement (see Section 3). We further demonstrate experimentally that the magnitude of $\tau_{\text{Jp}}$ correlates with weight disentanglement error and normalized accuracy (see Figures 1 and 2). This discussion provides a necessary condition for weight disentanglement and shows, both theoretically and experimentally, that explicitly introducing a regularization term to constrain $\tau_{\text{Jp}}$ during fine-tuning improves weight disentanglement.
>
> Our contribution lies not only in achieving performance improvements through regularization but also in unveiling the causal mechanism underlying task arithmetic and weight disentanglement. This work extends the methodological discussion in a way that enhances the understanding of these mechanisms and their implications.

---

> > ### Comment · Reviewer_n6ZH · 2024-12-01
> >
> > Thank you for your response and for providing further clarification.
> >
> > I appreciate the effort and substantial work the authors have put into this paper. However, after carefully reviewing the manuscript, I still find the contributions incremental compared to [1]. While the paper adds some theoretical and experimental insights, it appears to primarily build upon existing ideas without sufficiently distinguishing itself.
> >
> > The proposed regularization term is an interesting addition, but its novelty seems limited given the foundational similarity to [1], which already highlights the importance of model linearization and task-specific kernel localization. Despite the authors’ theoretical expansion and experimental validation, I remain unconvinced that this paper represents a significant advancement over prior work.
> >
> > That said, I acknowledge the effort involved in conducting this research and the methodological rigor in its presentation. After consideration, I maintain my borderline reject decision, as I believe further refinement is needed to emphasize the originality and impact of this work.

---

> ### Author Response · Authors · 2024-11-22
> **Response to reviewer n6ZH**
>
> ### Regarding Weakness 2
>
> >While the empirical results are compelling, the paper lacks a thorough theoretical explanation for why the proposed regularization leads to better performance compared to other methods, such as those discussed in Ortiz-Jimenez et al. (2023). I am confused about why a simple and soft regularization results in such improvement compared to [1]. A deeper theoretical analysis could strengthen the paper's contributions.
>
> We appreciate your feedback on the effectiveness of the proposed regularization.
>
> Our regularization term is defined as $||(\theta - \theta_0)^T \nabla_\theta f(\theta_0, x_{\text{other}})||^2 $, designed to encourage the task vector ($\theta - \theta_0$) to be orthogonal to the output gradient of other tasks $\nabla_\theta f(\theta_0, x_{\text{other}})$ in the pre-trained model (see Section 4.1 for details). Similar approaches, which aim to orthogonalize model weights to a specific vector by incorporating the L2 norm of their inner product as a regularization term, have been explored in previous studies (e.g., [2]) and have demonstrated their effectiveness. Our method is based on a similar idea, and our experimental results confirm that the task vector is effectively guided to be orthogonal to $\nabla_\theta f(\theta_0, x_{\text{other}})$. This effect, as a result, keeps $ \tau_{\text{Jp}}$ small, and the relationship between this reduction, improved weight disentanglement, and enhanced task arithmetic performance is elaborated in Section 3.
>
> [2] Wang, Xiao, et al. "Orthogonal subspace learning for language model continual learning." arXiv preprint arXiv:2310.14152 (2023).

---

> ### Author Response · Authors · 2024-11-22
> **Response to reviewer n6ZH**
>
> ### Regarding Weakness 3
>
> >The authors briefly mention tuning the regularization strength but do not provide sufficient details on how this hyperparameter was selected. The sensitive analysis of this hyperparameter is also necessary for the paper.
>
> We thank you for your comments regarding the hyperparameter $\lambda$.
>
> The strength of the regularization term, $\lambda$, was tuned through a grid search over [1e-3, 1e-2, 1e-1], using validation accuracy as the evaluation metric. Due to limited computational resources, we reused the $\lambda$ value obtained from a specific task (Image: Cars, NLP: CoLA, Civil Comments) across all experiments. While further analysis of the sensitivity to $\lambda$ is necessary, we empirically confirmed that using a unified $\lambda$ across all experiments still yields the benefits of the proposed regularization, suggesting that $\lambda$ is not overly sensitive.

---

> ### Author Response · Authors · 2024-11-22
> **Response to reviewer n6ZH**
>
> ### Regarding the Question
>
> >Could the proposed regularization affect the model's plasticity? Specifically, how might the addition of this regularization impact the fine-tuning performance, potentially influenced by the strength of the regularization?
>
> We appreciate your question regarding the impact of our regularization on fine-tuning performance.
>
> The impact of regularization on fine-tuning performance is shown in Figure 4 in the Appendix E.1. Compared to Linear FT (before regularization), our method (Ours) maintains comparable performance. The performance gap between Linear FT (including Ours) and Non-linear FT arises from the *non-linear advantage* [3], where fine-tuning in the non-linear regime achieves higher performance due to the richer expressivity of the non-linear loss landscape.
>
> [3] Fort, Stanislav, et al. “Deep learning versus kernel learning: an empirical study of loss landscape geometry and the time evolution of the neural tangent kernel.” *Advances in Neural Information Processing Systems* 33 (2020): 5850-5861.

---

> ### Author Response · Authors · 2024-11-29
> **Response to reviewer n6ZH**
>
> We sincerely appreciate the time you have dedicated to reviewing our work. We are fully committed to revising our paper in response to your feedback, so please feel free to share any additional concerns or suggestions.

---

> ### Author Response · Authors · 2024-12-01
> **Regarding the novelty of our work**
>
> We appreciate the time you have taken to review our paper.
>
> Thank you for pointing out your concerns regarding the novelty of our work.
> To address your feedback, we outline below the differences between our work and [1], as well as the key novelties:
>
> ### **1. A methodologically extensible interpretation of Weight Disentanglement (Sec. 3)**
>
> First, we clarified the conditions required for weight disentanglement through a theoretically extensible discussion. As mentioned previously, [1] associates task-specific kernel localization with weight disentanglement; however, the causal considerations and practical means of achieving task-specific kernel localization remained unresolved. This problem had remained unexplored, but we theoretically demonstrated that the causal factor is $\tau \text{Jp}$, and we also provided experimental evidence to substantiate this claim.
>
> ### **2. Proposal of an Efficient Method Achieving Significant Performance Improvements (Sec. 4.1 and 4.2)**
>
> Based on the discussion in Novelty 1, we proposed a novel regularization method as a practical approach. This method is efficient in terms of both computational and memory costs, yet achieves up to a 10.2-point improvement in accuracy over [1]’s Linear FT in task addition.
>
> The efficiency of our method is demonstrated in the table below, which shows the costs of fine-tuning, task vector coefficient tuning during merging, and their total. The additional cost of our method compared to Linear FT during fine-tuning is kept minimal. Furthermore, our method achieves significant performance improvements even without task vector coefficient tuning. As a result, the total cost of our method is lower than that of Linear FT while achieving much higher performance.
>
> We recognize that this improvement is not merely incremental but represents a significant and meaningful contribution.
>
> | Method | Fine-tuning |  | Merging |  | Total | Accuracy  |
> | --- | --- | --- | --- | --- | --- | --- |
> |  | Time (min) | Memory / device (GB) (4 GPU used) | Time (min) | Memory (GB) (1 GPU used) | Time (min) |  |
> | Non-lin. FT | 57 | 4.5 | 18 | 1.5 | 75 | 70.4 |
> | Linear FT | 96 | 6.2 | 37 | 4.8  | 133 | 74.3 |
> | Ours(without coef. tuning) | 100 | 6.4 | 0 | 0 | 100 | 84.2 |
> | Ours(with coef. tuning) | 100 | 6.4 | 37 | 4.8 | 137 | **84.5** |
>
> ### **3. Enhanced Practical Feasibility (Sec. 4.3)**
>
> In Section 4.3, we demonstrated the performance of our method under more constrained scenarios, where significant performance gains over the method in [1] were still observed. Moreover, in the context of Domain Adaptation, methodologies that leverage unlabeled data from the target domain to improve target performance, known as Unsupervised Domain Adaptation (UDA) [2], have been extensively studied. Unlike [1] and other conventional approaches, our work extends UDA methodologies by utilizing unlabeled data from other tasks for the purpose of weight disentanglement. We consider this extension to be another novelty of our study.
>
> **In summary**, the above novelties are not minor incremental improvements over [1], but rather substantial mitigations of bottlenecks in the practical application of task arithmetic. We believe our work should be evaluated accordingly.
>
> We would greatly appreciate it if you could reconsider your score in light of these points. If you have any additional concerns, please do not hesitate to let us know.
>
> [2] Ganin, Yaroslav, and Victor Lempitsky. “Unsupervised domain adaptation by backpropagation.” International Conference on Machine Learning. PMLR, 2015.

---

### Official Review · Reviewer_rGLy · 2024-10-30

**Soundness:** 3
**Presentation:** 3
**Contribution:** 3
**Rating:** 6
**Confidence:** 2

**Summary:**

The paper introduces a novel metric, $\tau \text{Jp}$ ($\tau$-Jacobian product), to improve understanding of weight disentanglement in task arithmetic. It demonstrates that τJp inversely correlates with normalized accuracy, suggesting it as an indicator for weight disentanglement. A regularization technique is proposed to minimize τJp during fine-tuning, effectively reducing the need for coefficient adjustments in task addition and negation. It also proves valuable in incremental learning scenarios where future tasks are unknown.

**Strengths:**

1.The paper addresses an important and timely topic: in an era where foundation models are prevalent, better understanding weight disentanglement is particularly valuable for enhancing the practical applicability of these models.

2.The proposed metric offers a deeper understanding of weight disentanglement, and the regularization method effectively reduces task interference, minimizing the need for coefficient adjustments.

3.The success of the proposed method in incremental learning scenarios aligns well with real-world applications, demonstrating its scalability and practical relevance when future tasks are unknown.

**Weaknesses:**

1.While the paper introduces the $\tau \text{Jp}$ metric and explains its relationship with weight disentanglement, the theoretical justification for why $\tau \text{Jp}$ regularization effectively reduces task interference could be further elaborated.

2.The proposed regularization method lacks a comparison with other existing regularization techniques, which makes it difficult to fully assess its relative strengths and weaknesses.

3.The paper mentions task addition, task negation, and task analogies in the introduction and background sections as key operations in task arithmetic, but there are no experiments evaluating task analogies. This inconsistency weakens the completeness of the experimental validation.

**Questions:**

Suggestions：

1.The related work section could be improved by explicitly connecting prior studies to this paper's contributions, emphasizing how the proposed method addresses existing limitations.
2.Consider moving the related work section after the methods section, especially since the current structure delays the introduction of the proposed method until page 5. This change would allow readers to quickly understand the proposed approach before diving into comparisons, enhancing readability and engagement.

---

> ### Author Response · Authors · 2024-11-22
> **Response to reviewer rGLy**
>
> ### Regarding Weakness 1
> >While the paper introduces the $\tau Jp$ metric and explains its relationship with weight disentanglement, the theoretical justification for why  $\tau Jp$ regularization effectively reduces task interference could be further elaborated.
>
> Thank you for your insightful feedback.
>
> Our regularization term is defined as $||(\theta - \theta_0)^T \nabla_\theta f(\theta_0, x_{\text{other}})||^2 $, designed to encourage the task vector ($\theta - \theta_0$) to be orthogonal to the output gradient of other tasks $\nabla_\theta f(\theta_0, x_{\text{other}})$ in the pre-trained model (see Section 4.1 for details). Similar approaches, which aim to orthogonalize model weights to a specific vector by incorporating the L2 norm of their inner product as a regularization term, have been explored in previous studies (e.g., [1]) and have demonstrated their effectiveness. Our method is based on a similar idea, and our experimental results confirm that the task vector is effectively guided to be orthogonal to $\nabla_\theta f(\theta_0, x_{\text{other}})$. This effect, as a result, keeps $ \tau_{\text{Jp}}$, and the relationship between this reduction, improved weight disentanglement, and enhanced task arithmetic performance is elaborated in Section 3.
>
> [1] Wang, Xiao, et al. "Orthogonal subspace learning for language model continual learning." arXiv preprint arXiv:2310.14152 (2023).

---

> ### Author Response · Authors · 2024-11-22
> **Response to reviewer rGLy**
>
> ### Regarding Weakness 2
>
> >The proposed regularization method lacks a comparison with other existing regularization techniques, which makes it difficult to fully assess its relative strengths and weaknesses.
>
> Thank you for your suggestions to include additional methods and revise the tables.
>
> In response, we conducted experiments with other existing task arithmetic methods and compared them. Specifically, we added TIES-Merging and AdaMerging to Table 1 for direct comparison. Regarding AdaMerging, its training process for task vector coefficients required significant GPU memory, making it infeasible to implement for all model sizes within our constrained environment. Therefore, we reported its results based on those provided by the original authors [2] under the same experimental settings. Additionally, we included TIES-Merging in other experiments (Tables 2–4), including NLP tasks as well as image tasks, as shown in Tables 3 and 4. Our method outperformed both TIES-Merging and AdaMerging in all cases.
>
> [2] Yang, Enneng, et al. "Adamerging: Adaptive model merging for multi-task learning." arXiv preprint arXiv:2310.02575 (2023).

---

> ### Author Response · Authors · 2024-11-22
> **Response to reviewer rGLy**
>
> ### Regarding Weakness 3
>
> >The paper mentions task addition, task negation, and task analogies in the introduction and background sections as key operations in task arithmetic, but there are no experiments evaluating task analogies. This inconsistency weakens the completeness of the experimental validation.
>
> Thank you for your thoughtful insights regarding task analogies.
>
> This study is based on the task arithmetic defined in Equation 1 of [3]. As this definition does not account for task analogies, we limited task arithmetic in this work to addition and negation only. To avoid confusion, we have removed references to task analogies from the explanation of task arithmetic and instead added the above explanation as a footnote on page 2 to clarify this point.
>
> [3] Guillermo Ortiz-Jimenez, Alessandro Favero, and Pascal Frossard. Task arithmetic in the tangent space: Improved editing of pre-trained models. Advances in Neural Information Processing Systems, 2023.

---

> ### Author Response · Authors · 2024-11-22
> **Response to reviewer rGLy**
>
> ### Regarding Questions
>
> >1.The related work section could be improved by explicitly connecting prior studies to this paper's contributions, emphasizing how the proposed method addresses existing limitations.
>
> >2.Consider moving the related work section after the methods section, especially since the current structure delays the introduction of the proposed method until page 5. This change would allow readers to quickly understand the proposed approach before diving into comparisons, enhancing readability and engagement.
>
> Thank you for your valuable suggestions.
>
> First, we have improved the Related Work section to clearly articulate the connection between existing methods and our proposed approach. Specifically, we highlighted the limitations of existing methods and demonstrated how our work addresses these issues. Additionally, we included discussions on related studies concerning the idea of $\tau_{\text{Jp}}$ regularization, providing a more comprehensive context for our contributions.
>
> In addition, we have relocated the Related Work section to follow the Method section for improved logical flow and clarity.

---

> > ### Comment · Reviewer_rGLy · 2024-11-22
> >
> > Thanks for the detailed responses and revisions—they address my concerns well, so I’ve updated my score!

---

> > > ### Author Response · Authors · 2024-11-23
> > > **Response to reviewer rGLy**
> > >
> > > Thank you very much for raising the score and for your thoughtful review. We believe we have addressed all of your concerns, but please let us know if you have any further questions or additional concerns.

---

### Official Review · Reviewer_onTx · 2024-10-31

**Soundness:** 2
**Presentation:** 3
**Contribution:** 3
**Rating:** 5
**Confidence:** 5

**Summary:**

This paper proposes a new metric called $\tau$Jp ($\tau$-Jacobian product) for measuring weight disentanglement in task arithmetic operations on neural networks. The authors theoretically analyze the relationship between $\tau$Jp and interference between tasks, and introduce a regularization method based on minimizing $\tau$Jp during fine-tuning. Experiments on image classification tasks demonstrate improved performance and reduced need for hyperparameter tuning compared to existing methods.

**Strengths:**

1. The paper provides a comprehensive theoretical and empirical study of the relationship between the proposed $\tau$Jp metric and task interference in neural networks.

2. The introduction of $\tau$Jp as a new metric for weight disentanglement is novel and well-motivated.

3. The proposed regularization method eliminates the need for tuning inference-time hyperparameters ($\alpha$), which is a practical advantage.

**Weaknesses:**

1. The method requires access to data from all other tasks during training, which is often unavailable in realistic task arithmetic scenarios. This limits the practical applicability of the approach.

2. The computational cost of calculating τJp is likely very high, as it involves multiple Jacobian-vector products. The paper does not report runtime or resource requirements, making it difficult to assess scalability.

3. Experiments are limited to image classification tasks. Evaluation on other domains like language tasks would strengthen the claims of generality.

4. The derivation of Equation 7 from the weight disentanglement definition is non-trivial and should be explained more clearly.

**Questions:**

- How does the computational cost of the proposed method compare to existing approaches?
- Can the method be adapted to work with limited or no access to data from other tasks?
- How well does the approach generalize to other domains beyond image classification?

---

> ### Author Response · Authors · 2024-11-22
> **Response to reviewer onTx**
>
> ### Regarding Weakness 1 and Question 2
> >The method requires access to data from all other tasks during training, which is often unavailable in realistic task arithmetic scenarios. This limits the practical applicability of the approach.
>
> >Can the method be adapted to work with limited or no access to data from other tasks?
>
> Thank you for raising this important concern and for your thoughtful question.
>
> Our method does indeed require access to datasets from other tasks; however, we clarify that, in the case of classification tasks, only access to unlabeled data is necessary (see Table 7). This naturally extends the context of Unsupervised Domain Adaptation (UDA) [1], where access is limited to the data distribution x of other domains. By utilizing this access, our method learns task vectors in orthogonal directions to $\nabla_{\theta}f(x, \theta_0)$ of other tasks, enhancing model weight disentanglement. Unlike other methods, which cannot effectively utilize the data distribution x in these scenarios, our approach provides an effective learning strategy under these conditions.
>
> Furthermore, in Section 4.3, we demonstrate that our method remains effective even in scenarios where access is limited to specific tasks or where the amount of accessible data from other tasks is constrained. Even minimal regularization-based learning with a few steps significantly improves task arithmetic performance, highlighting the robustness of our approach under these conditions.
>
> [1] Ganin, Yaroslav, and Victor Lempitsky. "Unsupervised domain adaptation by backpropagation." International conference on machine learning. PMLR, 2015.

---

> ### Author Response · Authors · 2024-11-22
> **Response to reviewer onTx**
>
> ### Regarding Weakness 2 and Question 1
> >The computational cost of calculating τJp is likely very high, as it involves multiple Jacobian-vector products. The paper does not report runtime or resource requirements, making it difficult to assess scalability.
>
> >How does the computational cost of the proposed method compare to existing approaches?
>
> Thank you for your insightful comments. Below, we provide details regarding the computational complexity, runtime, and required resources for our proposed method.
>
> First, as you correctly pointed out, the computation of the regularization term involves Jacobian calculations. However, this can be efficiently performed using Jacobian-vector products (JvPs), which are generally computed with the same complexity as a forward pass through forward-mode automatic differentiation [2]. To further improve computational efficiency, we used a batch size of  $\frac{1}{8}$ of the batch size used for computing the loss on the target task for vision tasks, and  $\frac{1}{4}$ for NLP tasks. This ensures that the computation of the regularization term has minimal impact on the overall computational cost of the forward pass.
>
> The actual runtime and corresponding performance results are shown in Table 6 of Appendix C. We have also included a baseline for the updated results, which corresponds to the case without any regularization (No reg., i.e., Linear FT). While computing the regularization term for all other tasks at every iteration (as in Eq. 10) is highly effective, it comes with a significant computational cost. To address this limitation, we introduced cyclical regularization, where only one task is considered for regularization at each iteration, and tasks are cycled through. This approach successfully reduces runtime by approximately 80% while maintaining comparable performance. The runtime for cyclical regularization is nearly identical to that of Linear FT (no regularization) while achieving substantial performance improvements. Furthermore, the runtime of cyclical regularization does not depend on the number of tasks being considered, demonstrating its scalability.
>
> Additionally, as discussed in [3], linearizing fine-tuning generally incurs approximately 2–3 times the computational cost compared to traditional Non-lin. FT. This is a broader challenge within the context of linear fine-tuning, not limited to our method. However, as outlined in Section 6, approaches such as LoRA and other parameter-efficient techniques have been shown to reduce the computational cost of linearized fine-tuning. This opens opportunities for further improving the efficiency of our method. Whether our regularization approach retains its effectiveness when combined with such efficient methods remains an open question and is left as future work.
>
> Based on these findings, we conclude that our proposed method operates within a scalable computational budget and is practical for implementation and experimentation with realistic runtime and resource requirements.
>
> [2] Baydin, Atilim Gunes, et al. “Automatic differentiation in machine learning: a survey.” *Journal of Machine Learning Research* 18.153 (2018): 1-43.
>
> [3] Guillermo Ortiz-Jimenez, Alessandro Favero, and Pascal Frossard. Task arithmetic in the tangent space: Improved editing of pre-trained models. Advances in Neural Information Processing Systems, 2023.

---

> > ### Comment · Reviewer_onTx · 2024-11-22
> > **Reply to Authors**
> >
> > Hi, could you provide some results regarding computational cost?

---

> ### Author Response · Authors · 2024-11-22
> **Response to reviewer onTx**
>
> ### Regarding Weakness 3 and Question 3
> >Experiments are limited to image classification tasks. Evaluation on other domains like language tasks would strengthen the claims of generality.
>
> >How well does the approach generalize to other domains beyond image classification?
>
> Thank you for your valuable feedback. We have conducted additional experiments in the NLP domain, with the results summarized in Tables 3, 4, 8, and 9.
>
> First, we performed task addition experiments (Table 3 and 8) on four tasks from the GLUE benchmark using the T5 model. The selected tasks followed the setup in [4]. Incorporating regularization consistently improved both absolute and normalized accuracy. Notably, for the SST-2 task, adding tasks without regularization resulted in significant performance degradation (normalized accuracy: 61.3). This phenomenon suggests potential interference between task vectors, particularly from CoLA, another single-sentence task. Our proposed regularization substantially mitigated this issue, improving the normalized accuracy to 98.5.
>
> Additionally, we conducted task negation experiments (Table 4 and 9) using GPT-2 to achieve less toxic text generation. In this setup, we performed causal language modeling on toxic texts (Civil Comments) and subtracted the resulting task vector from a pre-trained model. Our method achieved the most effective toxicity reduction while retaining the model’s original language capabilities, as measured by perplexity on WikiText-103. In contrast, other methods exhibited a trade-off between toxicity reduction and language capability retention due to interference from the task vector. Our approach significantly alleviates this trade-off (See Tables 9).
>
> [4] Ilharco, Gabriel, Marco Tulio Ribeiro, Mitchell Wortsman, Suchin Gururangan, Ludwig Schmidt, Hannaneh Hajishirzi, and Ali Farhadi. “Editing models with task arithmetic.” arXiv preprint arXiv:2212.04089 (2022).

---

> ### Author Response · Authors · 2024-11-22
> **Response to reviewer onTx**
>
> ### Regarding Weakness 4
> >The derivation of Equation 7 from the weight disentanglement definition is non-trivial and should be explained more clearly.
>
> We greatly appreciate your helpful feedback.
>
> First, regarding the definition of weight disentanglement, we kindly refer you to the beginning of the second paragraph in Section 2.2 for a detailed explanation.
>
> Next, we have added further clarification on how Equation 7 is derived from this definition of weight disentanglement (highlighted in red in Section 3.1). Specifically, satisfying weight disentanglement means that:
>
> $f\left(x_A ; \theta_0+\alpha_A \tau_A+\alpha_B \tau_B\right)= f\left(x_A ; \theta_0+\alpha_A \tau_A\right)$
>
> and
>
> $f\left(x_B ; \theta_0+\alpha_A \tau_A+\alpha_B \tau_B\right) = f\left(x_B ; \theta_0+\alpha_B \tau_B\right)$
>
> for any $\alpha_A$ and $\alpha_B$. Based on Equation 6, this naturally leads to the derivation of Equation 7.

---

> ### Author Response · Authors · 2024-11-24
> **Reply to reviewer onTx**
>
> ### Regarding Computational Cost
>
> >Hi, could you provide some results regarding computational cost?
>
> Thank you for your prompt response. Regarding the computational cost of our method, we would like to address your concern.
>
> As mentioned in our previous reply under “Regarding Weakness 2 and Question 1,” we provided a detailed explanation including the runtime per iteration (Sec. / Iter.) and included results in Table 6 of Appendix C. For clarity, we present the table below:
>
> | Method | Abs. (↑) | Norm. (↑) | Sec. / Iter. (↓) |
> | ---- | :---:   | :---:  | :---:  |
> | No reg. (Linear FT) | 74.3 |85.0 |0.361 |
> | Ours (Eq. 11) | 84.5 |97.6 |0.374 |
>
> These results are based on the task addition experiment using ViT-B-32 shown in Table 1. The runtime increase introduced by our regularization term is approximately 0.01 seconds per iteration, which is minimal.
>
> We hope this addresses your concern and clarifies our approach. Thank you again for your valuable feedback and for helping us improve the quality of our paper. If you have any additional questions or concerns, please feel free to let us know.

---

> > ### Comment · Reviewer_onTx · 2024-11-25
> > **Further question**
> >
> > Thanks for the detailed reply, here are some further questions:
> > - While additional data are unlabeled, seeing test samples seems unfair compared to other baselines except for Adamerging; Could you provide some comparisons to other baselines, e.g. traditional MTL baselines which have access to all test samples?
> > - Could you provide results regarding memory consumption in addition to iteration time?

---

> > > ### Author Response · Authors · 2024-11-26
> > > **Reply to reviewer onTx**
> > >
> > > We appreciate your additional questions, which contribute to improving the quality of our paper.
> > >
> > > ### **Regarding Fair Comparison**
> > >
> > > > While additional data are unlabeled, seeing test samples seems unfair compared to other baselines except for Adamerging; Could you provide some comparisons to other baselines, e.g. traditional MTL baselines which have access to all test samples?
> > >
> > > First, we clarify that when using unlabeled data from other tasks during fine-tuning, we rely on training data, not test data. Therefore, the data used during fine-tuning does not directly overlap with the data used for evaluation.
> > >
> > > Nonetheless, our method differs from other fine-tuning approaches, such as Non-linear FT and Linear FT, in the range of accessible data during fine-tuning. To address this, we have already included the results of traditional MTL (”MTL”) in Table 1. While MTL achieves very high absolute accuracy by leveraging labeled data from all tasks, our method demonstrates comparable performance despite being restricted to accessing only unlabeled data.
> > >
> > > Related to this discussion, we provide additional insights on data accessibility during fine-tuning in Appendix D (see especially Table 7). Briefly, while MTL achieves high multi-task performance, it requires labeled data for all tasks during fine-tuning and lacks the flexibility to add new capabilities or modify existing ones without forgetting prior knowledge. In contrast, our method retains the inherent flexibility of task arithmetic while utilizing unlabeled data from other tasks during fine-tuning, achieving performance comparable to MTL.
> > >
> > > ### **Regarding Memory Consumption**
> > >
> > > > Could you provide results regarding memory consumption in addition to iteration time?
> > >
> > > Below, we present the results of the GPU memory consumption increase caused by the addition of our regularization term in the experiments from Table 1, broken down by model size. The peak size of allocated memory is reported in gigabytes (GB). As stated in the paper, gradient accumulation is used for ViT-L-14. As previously demonstrated in our response, the efficient implementation of the regularization ensures that the increase in memory consumption is kept minimal.
> > >
> > > | Method                 |  ViT-B-32  |  ViT-B-16  |  ViT-L-14  |
> > > | :--------------------- | :--------: | :--------: | :--------: |
> > > | No reg. (Linear FT)     |    6.18    |   13.38    |   15.00    |
> > > | Ours                    |    6.38    |   13.69    |   15.71    |
> > > | **Increase**                | 0.20 (+3.2%) | 0.31 (+2.3%) | 0.71 (+4.7%) |
> > >
> > > We have addressed all your questions and also improved the presentation for those responses that may have been overlooked. We would greatly appreciate it if these responses could be taken into account when updating your score. Of course, if you have any further questions, please do not hesitate to ask.

---

> > > > ### Comment · Reviewer_onTx · 2024-11-26
> > > > **Further question**
> > > >
> > > > Thanks for the reply, could you please provide the computational cost of both efficient and strict reg so that I could have a clear view? **I double-checked the code online and have the following question: in the penalty iter, the newly introduced data batch and additional jvp product both increase the memory consumption, I suspect nearly double memory compared to Linear FT. Besides, the authors deployed four V100 GPUs which have 64GB VRAM in total, why do the authors leave most memory idle?** Please correct me if I'm wrong.
> > > >
> > > > As stated before, my major concern is the unfair comparison with baselines. Existing model merging techniques can be broadly categorized into two main types (Yang et al., 2024): (i) Pre-Merging Methods: These methods focus on enhancing the conditions necessary for effective model merging by optimizing the fine-tuning process of individual models. (ii) During Merging Methods: These approaches address task conflicts and interference through various strategies before executing the parameter merging operations.
> > > >
> > > > The proposed method focuses on both fine-tuning and task conflicts, as a result, all training data across different tasks and additional computational resources are needed. While I do see clear positives in the paper, especially when compared to traditional task merging, I am still on the fence about the novelty/strength of the contribution and where exactly to place it in the literature. **Furthermore, it has a bottleneck while the model's size is increasing (e.g. ViT-L-14), I suppose it's because of the performance drop on single-task due to the reg.**
> > > >
> > > > Meanwhile, I'd like to consider this method as an MTL method, thus some results regarding generalization would strengthen this work, e.g. generalizing to an entirely unseen test set (table 3 from Adamerging). I encourage the authors to include this experiment if time permits.
> > > >
> > > > Currently, I am inclined to keep my rating. **I'm open to reconsidering my score if all the above concerns are addressed.**
> > > >
> > > > Yang et al. Model Merging in LLMs, MLLMs, and Beyond: Methods, Theories, Applications and Opportunities.
> > > >
> > > > Yang et al. AdaMerging: Adaptive Model Merging for Multi-Task Learning. ICLR 2024.

---

> > > > > ### Author Response · Authors · 2024-11-27
> > > > > **Reply to reviewer onTx**
> > > > >
> > > > > Thank you for your detailed response.
> > > > >
> > > > > ### **Regarding Memory Consumption**
> > > > > > could you please provide the computational cost of both efficient and strict reg so that I could have a clear view? I double-checked the code online and have the following question: in the penalty iter, the newly introduced data batch and additional jvp product both increase the memory consumption, I suspect nearly double memory compared to Linear FT. Besides, the authors deployed four V100 GPUs which have 64GB VRAM in total, why do the authors leave most memory idle? Please correct me if I'm wrong.
> > > > >
> > > > > First, below we present a comparison of runtime and memory consumption on ViT-B-32, including the strict regularization.
> > > > > The memory consumption values we previously reported represent the peak memory usage on a single device among the four V100 GPUs used in our setup. We have reported the highest memory usage among the four devices, as also indicated in the “Allocated / Device (GB)” column of the table below.
> > > > >
> > > > > |  | Abs. | Norm. | Sec. / Iter. | Allocated / Device (GB) |
> > > > > | :--- | :---: | :---: | :---: | :---: |
> > > > > | No reg. (Linear FT) | 74.3 | 85.0 | 0.361 | 6.18 |
> > > > > | Efficient reg. | 84.5 | 97.6 | 0.374 | 6.38 |
> > > > > | Strict reg. | 86.4 | 99.3 | 2.027 | 8.28 |
> > > > >
> > > > >
> > > > > As you correctly pointed out, the computation of the regularization term requires additional data batches and JvP calculations. However, as mentioned previously, the batch size used for calculating the regularization term is reduced to 1/8 of the original batch size used for loss computation in image tasks. This reduction ensures that the additional data batch and JvP computations do not dominate memory consumption. Consequently, the increase in memory usage is effectively minimized.

---

> > > > > ### Author Response · Authors · 2024-11-27
> > > > > **Reply to reviewer onTx**
> > > > >
> > > > > ### **Regarding Fair Comparison**
> > > > > >As stated before, my major concern is the unfair comparison with baselines. Existing model merging techniques can be broadly categorized into two main types (Yang et al., 2024): (i) Pre-Merging Methods: These methods focus on enhancing the conditions necessary for effective model merging by optimizing the fine-tuning process of individual models. (ii) During Merging Methods: These approaches address task conflicts and interference through various strategies before executing the parameter merging operations.
> > > > >
> > > > > >The proposed method focuses on both fine-tuning and task conflicts, as a result, all training data across different tasks and additional computational resources are needed. While I do see clear positives in the paper, especially when compared to traditional task merging, I am still on the fence about the novelty/strength of the contribution and where exactly to place it in the literature.
> > > > >
> > > > > As you correctly noted, our method primarily focuses on “fine-tuning” and suppressing “task conflicts.” However, this approach is based on linearized fine-tuning [1], which aligns with the context of **(i) Pre-Merging Methods**, rather than **(ii) During Merging Methods**. Specifically, while [1] belongs to (i), it emphasizes improving weight disentanglement—hence reducing task conflicts—through linearized fine-tuning, which is consistent with our objectives.
> > > > >
> > > > > Regarding the fairness of comparisons, we first emphasize that all existing model merging methods, including basic task arithmetic, require labeled data from all tasks during the adjustment of task vector coefficients. In contrast, our method demonstrates superior performance even when all coefficients are fixed at 1.0, without requiring coefficient adjustment. This ensures **fairness during evaluation**.
> > > > >
> > > > > However, as you pointed out, there is an inherent lack of fairness in terms of data accessibility during training, as different methods have different levels of access to data (see Table 7 for details). To address this, as suggested, we evaluated the performance of multi-task learning (MTL). This evaluation illustrates the impact of relaxed data accessibility constraints on task arithmetic performance and highlights the potential performance improvements achievable through our method’s use of unlabeled data.
> > > > >
> > > > > In summary, the contribution of our method lies in its novel approach within the context of **(i) Pre-Merging Methods**, effectively leveraging unlabeled data during training to significantly suppress task conflicts. This approach mitigates the costs and dependency on labeled data from all tasks required by conventional methods for coefficient adjustment.
> > > > >
> > > > > ### **Regarding Bottleneck of Our Method**
> > > > > > Furthermore, it has a bottleneck while the model's size is increasing (e.g. ViT-L-14), I suppose it's because of the performance drop on single-task due to the reg.
> > > > >
> > > > > Additionally, regarding your concerns about the bottlenecks associated with larger model sizes, it is true that as model size increases, the performance of other methods becomes comparable to ours (particularly with AdaMerging). First, addressing the concern that this is due to a decline in single-task performance caused by our regularization: as shown in Table 4 of Appendix E.1, the addition of our regularization results in little to no performance degradation for individual tasks compared to simple Linear FT (and in some cases, even improves performance).
> > > > >
> > > > > We believe the issue stems from the fact that as model size increases, Non-linear FT naturally becomes linearized, as demonstrated in Fig. 9 of [1], achieving sufficient linearization without requiring explicit linearized fine-tuning, which can inadvertently have a negative impact. For such large models, further performance improvements could potentially be achieved by using our regularization without explicit linearization.
> > > > >
> > > > > [1] Guillermo Ortiz-Jimenez, Alessandro Favero, and Pascal Frossard. Task arithmetic in the tangent space: Improved editing of pre-trained models. Advances in Neural Information Processing Systems, 2023.

---

> > > > > ### Author Response · Authors · 2024-11-27
> > > > > **Reply to reviewer onTx**
> > > > >
> > > > > ### **Regarding Generalization in the Context of MTL**
> > > > > >Meanwhile, I'd like to consider this method as an MTL method, thus some results regarding generalization would strengthen this work, e.g. generalizing to an entirely unseen test set (table 3 from Adamerging). I encourage the authors to include this experiment if time permits.
> > > > >
> > > > > As suggested, we evaluated the generalization performance of our method on unseen tasks within the context of MTL.
> > > > >
> > > > > First, we would like to emphasize that the primary objective of our method lies in **“weight disentanglement”**, as outlined in Equation 1 of the paper. Specifically, weight disentanglement aims to suppress interference between merged task vectors and minimize the negative impact on the pre-trained model’s original performance on tasks beyond the target. Thus, generalization to unseen tasks falls outside the direct scope of our method’s objectives. However, we also argue that weight disentanglement can be beneficial for generalization to unseen tasks when considered from the perspective of preserving the pre-trained model’s performance. We explain this further below.
> > > > >
> > > > > We conducted the experiments under the following setup:
> > > > >
> > > > > The training tasks consisted of six tasks—[Cars, GTSRB, DTD, EuroSAT, MNIST, SUN397]—while the unseen tasks were set as RESISC45 and SVHN, following the experimental setup in [2].
> > > > >
> > > > > Among the unseen tasks:
> > > > >
> > > > > •	**SVHN**: Similar to MNIST, SVHN is a 10-class digit classification task. If the knowledge learned from MNIST in the training tasks can be effectively leveraged, performance improvements on SVHN can be expected. Therefore, MTL’s generalization capabilities are likely more relevant than weight disentanglement for this task.
> > > > >
> > > > > •	**RESISC45**: Like EuroSAT, RESISC45 is a classification task for aerial imagery. However, RESISC45 includes 35 additional classes not covered by EuroSAT’s 10 classes. As such, the knowledge obtained from EuroSAT alone may not suffice for many instances in RESISC45. In this case, maintaining the pre-trained model’s knowledge through weight disentanglement is expected to result in higher performance.
> > > > >
> > > > > The experimental results, all measured in accuracy, are as follows:
> > > > >
> > > > > | Method | Training Tasks Avg. | SVHN | RESISC45 |
> > > > > | :--- | :---: | :---: | :---: |
> > > > > | Pre-trained | 48.8 | 31.6 | 60.2 |
> > > > > | Non-lin. FT | 73.4 | 50.2 | 52.2 |
> > > > > | Linear FT | 77.4 | 38.7 | 46.6 |
> > > > > | AdaMerging | 80.3 | **60.9** | 50.2 |
> > > > > | MTL | **86.3** | 60.8 | 42.9 |
> > > > > | Ours | 85.4 | 42.4 | **54.3** |
> > > > >
> > > > > As noted above, methods like MTL and AdaMerging, which do not prioritize weight disentanglement, demonstrate high generalization performance on SVHN. However, their performance on RESISC45 is significantly degraded, likely due to the negative impact on the pre-trained model’s knowledge. In contrast, our method, which focuses on weight disentanglement, maintains the pre-trained model’s performance on RESISC45 while showing lower generalization performance on SVHN. To reiterate, the reduced generalization performance on SVHN is a consequence of weight disentanglement and aligns with its definition in our approach.
> > > > >
> > > > > This discussion has been added to Appendix E.5 for further reference.
> > > > >
> > > > > [2] Yang et al. AdaMerging: Adaptive Model Merging for Multi-Task Learning. ICLR 2024.

---

> > > > > ### Author Response · Authors · 2024-11-27
> > > > > **Reply to reviewer onTx**
> > > > >
> > > > > We believe the above discussion addresses all of your concerns. We hope this will contribute to an updated evaluation of your score.

---

> > > > > > ### Comment · Reviewer_onTx · 2024-11-28
> > > > > > **Updated**
> > > > > >
> > > > > > Hi, thanks for your detailed responses, I'd like to update my score to a marginal score due to some limitations of the proposed method. I'm open to discussing this further with other reviewers and AC.
> > > > > >
> > > > > > - The proposed method is more like a pre-merging method, however, it requires additional training data compared to baselines.
> > > > > >
> > > > > > - While the superior performance has been achieved, we cannot neglect the additional computational cost and its bottleneck regarding model size. A more generalized and efficient method would be a benefit for the society.

---

> > > > > > > ### Author Response · Authors · 2024-11-29
> > > > > > > **Reply to reviewer onTx**
> > > > > > >
> > > > > > > We sincerely appreciate your score update and your openness to further discussions.
> > > > > > >
> > > > > > > We are confident that the limitations you highlighted regarding our method are either acceptable or do not constitute significant limitations. We address these points in the following comments and kindly encourage you to consider further updating your score. Additionally, we welcome further discussions on these topics involving other reviewers and the AC.

---

> > > > > > > ### Author Response · Authors · 2024-11-29
> > > > > > > **Further discussion 1**
> > > > > > >
> > > > > > > ### **The need for additional data during training**
> > > > > > >
> > > > > > > We acknowledge that, unlike other task arithmetic methods, our approach requires unlabeled data from other tasks during training, which could be perceived as a limitation.
> > > > > > >
> > > > > > > However, we argue that such a setting is often practical. Specifically, we view our approach as a natural extension of the Unsupervised Domain Adaptation (UDA) framework. In UDA, the input data distribution x from other domains is leveraged to improve in-domain performance, which aligns with the assumptions of our method. Through this, we address the issue of weight disentanglement—a problem that conventional pre-merging methods fail to solve—both theoretically and empirically.

---

> ### Author Response · Authors · 2024-11-29
> **Further discussion 2**
>
> ### **Computational and memory cost**
>
> As we have demonstrated, the increase in runtime and memory consumption caused by the addition of our regularization term during training is kept practical. Furthermore, during testing, our method achieves the highest performance without requiring coefficient tuning. In contrast, other methods rely on coefficient adjustment and access to labeled data from all tasks, leading to additional costs. We would like to emphasize that our method eliminates these costs and accessibility requirements at merging time.
>
> To provide a more detailed analysis, we present a comparison of runtime and memory consumption during Pre-merging and Merging stages in the table below. The runtime during Pre-merging was calculated based on the per-iteration runtime, while the Merging runtime for AdaMerging was taken from the authors’ reported results (note that a GeForce RTX 3090* was used in their experiments). While Linear FT and our method require linearized fine-tuning during Pre-merging, which takes approximately twice the time of standard fine-tuning, **the computational overhead introduced by our regularization (Ours - Linear FT) remains minimal at around 4%.** On the other hand, during the merging stage, other methods, particularly AdaMerging, incur significantly higher computational and memory costs. **In contrast, our method, especially without coefficient tuning, requires no merging costs, resulting in lower total costs while outperforming all other methods.**
>
> *GeForce RTX 3090, which has been reported to outperform the Tesla V100 we used in terms of FP32 FLOPS and other benchmarks, provides a significant advantage for AdaMerging in the comparison of the merging time. Despite this, our method is still faster.
>
> Additionally, as highlighted in Appendix D, task arithmetic is increasingly desired in applications such as multi-agent systems and personalized recommendation systems, where multiple task vectors can represent diverse models. For methods requiring coefficient tuning, the merging cost becomes increasingly dominant over pre-merging costs as the number of models to be represented grows. **In contrast, our method, which does not require coefficient tuning, maintains a constant total cost corresponding to the pre-merging stage, regardless of the number of models represented.** This indicates that our method offers highly efficient model editing in such scenarios.
>
> | Method                  | Pre-merging (Time min) | Pre-merging (Memory GB / device) | Merging (Time min) | Merging (Memory GB) | Total Time (min) | Accuracy (%) |
> |-------------------------|:------------------------:|:-----------------------------------:|:--------------------:|:---------------------:|:------------------:|:-------------:|
> | Non-lin. FT             | 57                    | 4.5                               | 18                 | 1.5                 | 75               | 70.4        |
> | Linear FT               | 96                    | 6.2                               | 37                 | 4.8                 | 133              | 74.3        |
> | AdaMerging              | 57                    | 4.5                               | 143                | >16 (OOM)*          | 200              | 80.1        |
> | Ours (w/o coef. tuning) | 100                   | 6.4                               | 0                  | 0                   | 100              | 84.2        |
> | Ours (w/ coef. tuning)  | 100                   | 6.4                               | 37                 | 4.8                 | 137              | **84.5**    |
>
> *OOM = Out of Memory on our device.
>
>
> Based on the arguments above, we are open to further discussions, involving other reviewers and the AC, regarding whether the computational and memory costs associated with our regularization term are significant or negligible. We welcome these discussions and are eager to provide additional insights.

---

> ### Author Response · Authors · 2024-11-29
> **Further discussion 3**
>
> ### **Model size bottleneck**
> | Method | Task vector coef. | ViT-B-32 Abs. (↑) | ViT-B-32 Norm. (↑) | ViT-B-16 Abs. (↓) | ViT-B-16 Norm. (↑) | ViT-L-14 Abs. (↓) | ViT-L-14 Norm. (↑) |
> | --- | --- | :---: | :---: | :---: | :---: | :---: | :---: |
> | Non-lin. FT | 1.0 | 19.9 | 20.5 | 19.1 | 19.7 | 37.6 | 39.0 |
> | Non-lin. FT | Grid-searched | 70.4 | 78.0 | 75.5 | 81.5 | 84.0 | 89.3 |
> | Linear FT  | 1.0 | 55.4 | 61.7 | 58.2 | 63.6 | 80.5 | 86.7 |
> | Linear FT | Grid-searched | 74.3 | 85.0 | 78.7 | 87.6 | 85.8 | 92.8 |
> | Ties-Merging ) | 1.0 | 74.2 | 84.8 | 78.6 | 87.6 | 85.0 | 91.9 |
> | Ties-Merging  | Grid-searched | 74.2 | 84.8 | 78.6 | 87.6 | 85.0 | 91.9 |
> | AdaMerging | Trained | 80.1 | 88.5 | 84.9 | 92.1 | **90.8** | 96.4 |
> | **Ours** | 1.0 | 84.2 | 97.2 | 87.5 | 98.4 | **90.8** | **99.0** |
> | **Ours** | Grid-searched | **84.5** | **97.6** | **87.6** | **98.5** | **90.8** | **99.0** |
>
> If by “bottleneck regarding model size,” you are referring to the diminishing improvements of our method over existing approaches as model size increases, we would like to highlight the following points:
>
> First, as previously noted, this phenomenon is not caused by any degradation in single-task performance due to the introduction of our regularization. This has been clearly demonstrated in Appendix E.1 (Table 4), where the addition of our regularization shows minimal or no degradation in single-task performance.
>
> The possible reasons for this phenomenon include the following:
>
> 1.	**Performance Saturation Observed Across Methods**
>
> The diminishing improvement margins from existing methods (e.g., Grid-searched Non-lin. FT) as model size increases are not unique to our approach. Similar behavior is observed in other existing methods, such as AdaMerging and TIES-Merging, and this is a well-known phenomenon in the ML community. For instance, when applying a novel method to train MNIST, significant improvements can be observed with a small model like LeNet-5, while the improvement margins are much smaller with a large model like ViT-Large. This is due to performance saturation, a common occurrence as models approach their capacity limits, and it is not specific to our method.
>
> 2.	**Natural Linearization of Non-lin. FT in Large Models**
>
> As model size increases, Non-lin. FT naturally becomes linearized, achieving sufficient linearization without requiring explicit linearized fine-tuning. In such cases, explicit linearization, which can sometimes involve performance degradation, becomes unnecessary and may even have adverse effects. For extremely large models, we anticipate further performance improvements by applying our regularization without relying on explicit linearized fine-tuning.

---

> > ### Comment · Reviewer_onTx · 2024-12-02
> > **Further discussion**
> >
> > Thanks for the detailed reply.
> >
> > Regarding the computational cost, I'm also a user of AdaMerging which I suspect has a comparable memory consumption with other methods. Could you kindly remind me where are the authors’ reported results?
> >
> > Regarding the bottleneck, it's natural to suspect that in a larger model (even larger than ViT-L-14), the proposed method could not beat baselines, since it achieved a 99% norm acc at the ViT-L-14 level. Thus, the additional data requirement and computational cost could not be neglected.

---

> ### Author Response · Authors · 2024-12-02
> **Reply to reviewer onTx**
>
> Thank you for your feedback.
>
> ### **Computational Cost of Our Method is Comparable to Other Methods**
>
> We directly implemented the AdaMerging code provided by the authors (they use NVIDIA RTX 3090 with 24GB of device memory) on their GitHub repository in our environment. However, on our single device (NVIDIA V100 with 16GB of device memory), the implementation resulted in an Out of Memory (OOM) error.
>
> From a runtime perspective, as reported by the authors in their paper, achieving sufficient performance with AdaMerging requires coefficient tuning, which adds an additional 125 minutes (even on higher-quality GPUs than ours) compared to conventional task arithmetic (see Tab. 12 of [2]). This makes the total runtime for AdaMerging approximately twice as long as our method.
>
> Based on the above observations, we are confident that the computational cost of our method is **comparable to, or even more efficient than, other methods**. Therefore, we do not consider it a weakness of our approach.
>
> ### **Model size bottleneck**
>
> >Furthermore, it has a bottleneck while the model's size is increasing (e.g. ViT-L-14), I suppose it's because of the performance drop on single-task due to the reg.
>
> First, as previously mentioned, we have already refuted your suggestion above. Adding our regularization does not significantly change single-task accuracy; in some cases, it even improves it.
>
> Second, **we respectfully disagree with the assessment** that our method has a bottleneck due to limited performance improvements in scenarios where large models already saturate performance. In such environments, there is inherently little room for improvement, and attributing this to a weakness of our method is not fair.
> For example, many widely accepted and commonly used techniques, such as regularization and data augmentation, which are considered broadly beneficial to communities, do not improve performance (or have no room for improvement) in such environments.
>
> Our approach remains **highly effective for applications with limited model sizes or more challenging tasks where performance is not saturated**. In these situations, our method achieves benchmarks that other methods do not reach. Even when the problem becomes easier due to high model expressiveness, incremental improvements are still valuable for many applications. While we agree that adding methods with additional computational costs may not be necessary in saturated settings, our method offers superior performance over existing techniques with less computational and memory overhead than alternatives like AdaMerging.
>
> In light of these considerations, we kindly request that you reconsider your confidence and score.
>
> [2] Yang et al. AdaMerging: Adaptive Model Merging for Multi-Task Learning. ICLR 2024.

---

### Official Review · Reviewer_hopm · 2024-11-03

**Soundness:** 3
**Presentation:** 3
**Contribution:** 3
**Rating:** 8
**Confidence:** 3

**Summary:**

The paper tackles the task of task arithmetics, i.e. how to combine *task vectors/parameters* to form a multi-task model. A key issue is to determine the best combination weights as to minimise interference between tasks, and maximise sharing of information / positive transfer.
More specifically, the authors make use of two previously introduced notions: **(i)** the notion of **weight disentanglement** which was  proposed as a measure of task interference in task arithmetic. And **(ii)** the Neural Tangent Kernel (NTK) which designates a training regime where parameter updates can be expressed with a linearised approximation.
Previous works have suggested that performing task arithmetics under the NTK regime can lead to better MTL performance. the authors investigate this behaviour in more depth. Based on this analysis, they also propose a regularisation technique to further reduce task interference when performing task arithmetic, which involves slightly fine-tuning the task vectors themselves.

**Strengths:**

* The paper is well motivated and grounded in previous related work
* The proposed method is simple and could adapt to different task arithmetic variants
* Interesting insights on the link between the proposed regularisation and weight disentanglement
* A more efficient implementation of the method is proposed for handling larger number of tasks (Equation 11)

**Weaknesses:**

* Missing baselines on more recent task arithmetic work: The main tables should include some recent task arithmetic results (e.g. TIES-Merging and AdaMerging) as well as standard single task and MTL baselines (although it is in the appendix), if only to better understand the existing gap in performance.
* Missing discussion about the extra cost: The paper briefly mentions efficiency of the method (e.g. equation 11 or line 364), however I think this could be discussed in more depth: On the one hand, the proposed method seems more robust to task coefficients $\alpha$, which could save on hyper parameter tuning; On the other hand, it involves a fine-tuning procedure which requires knowledge/access to all tasks simultaneously (Equation 10) as opposed to directly combining task vectors obtained independently from one another.

**Questions:**

* I find the notion of "One-shot" and "fine-tuned" experimental setting could be improved; First because the notion of fine-tuning can become confusing between the coefficients $\alpha$ vs the model parameters $\theta$ fine-tuning. Second, because it is not clear if it is referring to a specific method/objective for fine-tuning the task coefficients (e.g. AdaMerging or others) or simply hyperparameter search.

---

> ### Author Response · Authors · 2024-11-22
> **Response to reviewer hopm**
>
> ### Regarding Weakness 1
> > Missing baselines on more recent task arithmetic work: The main tables should include some recent task arithmetic results (e.g. TIES-Merging and AdaMerging) as well as standard single task and MTL baselines (although it is in the appendix), if only to better understand the existing gap in performance.
>
> Thank you for your valuable suggestions to include additional methods and revise the tables.
>
> In response, we conducted experiments with other existing task arithmetic methods and compared them. Specifically, we added TIES-Merging and AdaMerging to Table 1 for direct comparison. Regarding AdaMerging, its training process for task vector coefficients required significant GPU memory, making it infeasible to implement for all model sizes within our constrained environment. Therefore, we reported its results based on those provided by the original authors under the same experimental settings. Additionally, we included TIES-Merging in other experiments (Tables 2–4), including NLP tasks as well as image tasks, as shown in Tables 3 and 4. Our method outperformed both TIES-Merging and AdaMerging in all cases.
>
> Additionally, we incorporated the performance of MTL and standard single-task models (“Individual” in the tables) into the results.

---

> ### Author Response · Authors · 2024-11-22
> **Response to reviewer hopm**
>
> ### Regarding Weakness 2
>
> >Missing discussion about the extra cost: The paper briefly mentions efficiency of the method (e.g. equation 11 or line 364), however I think this could be discussed in more depth: On the one hand, the proposed method seems more robust to task coefficients
> , which could save on hyper parameter tuning; On the other hand, it involves a fine-tuning procedure which requires knowledge/access to all tasks simultaneously (Equation 10) as opposed to directly combining task vectors obtained independently from one another.
>
> We greatly appreciate your insightful and constructive feedback. Below, we address your points in detail.
>
> **Efficiency.**
> First, regarding the efficiency of our approach, the loss function shown in Equation 11 reduces the computational cost compared to Equation 10. Specifically, in Equation 10, the regularization term grows with the number of tasks considered at each iteration, whereas in Equation 11, the regularization is computed for only a single task at each iteration. This ensures that the computational cost per iteration remains constant, regardless of the number of tasks, making it a scalable method. As shown in Appendix C, we compared the runtime per iteration and found that Equation 11 achieves approximately an 80% reduction in runtime compared to Equation 10, while maintaining comparable performance. Furthermore, the runtime of Equation 11 is nearly identical to that of Linear FT (without regularization), while achieving significant performance improvements. These results demonstrate that Equation 11 is an efficient and highly effective approach.
>
> Regarding the tuning of task vector coefficients, as you correctly pointed out, our method achieves strong performance, particularly in task addition, without requiring coefficient tuning. This eliminates the need for manual adjustment. Traditional methods that rely on coefficient tuning suffer from increased computational costs as the number of tasks, dataset size, or parameter count grows, limiting their scalability. In contrast, our method excels in scalability by addressing this issue.
>
> **Access to Other Tasks During Fine-Tuning.**  Next, regarding the requirement for access to other tasks during fine-tuning, we clarify that while our method does require access to datasets from other tasks, in the case of classification tasks, it only needs access to unlabeled data (See Table 7). This aligns naturally with the context of Unsupervised Domain Adaptation (UDA) [1], where access to the data distribution x of other domains is leveraged. By utilizing this access, our method learns task vectors in orthogonal directions to $\nabla_{\theta}f(x, \theta_0)$ of other tasks, enhancing model weight disentanglement. Unlike other methods that cannot effectively utilize data distributions x under such scenarios, we propose a viable approach for learning in these conditions.
>
> Furthermore, in Section 4.3, we demonstrate that our method remains effective even in scenarios where access is limited to specific tasks or where the amount of accessible data from other tasks is constrained. Even minimal regularization-based learning with a few steps significantly improves task arithmetic performance, highlighting the robustness of our approach under these conditions.
>
> [1] Ganin, Y., & Lempitsky, V. (2015). Unsupervised domain adaptation by backpropagation. International Conference on Machine Learning (ICML). PMLR.

---

> ### Author Response · Authors · 2024-11-22
> **Response to reviewer hopm**
>
> ### Regarding the Question
> >I find the notion of "One-shot" and "fine-tuned" experimental setting could be improved; First because the notion of fine-tuning can become confusing between the coefficients $\alpha$ vs the model parameters $\theta$ fine-tuning. Second, because it is not clear if it is referring to a specific method/objective for fine-tuning the task coefficients (e.g. AdaMerging or others) or simply hyperparameter search.
>
> Thank you for your helpful suggestions. We agree with your suggestion and have updated the notation accordingly. Specifically, in Tables 1 and 2, we have changed the column title to “Task Vector Coef.” and revised the notation to indicate “1.0” for cases without coefficient tuning and “Grid-searched” for cases with tuning. We believe this clarification reduces potential confusion regarding the representation of coefficient adjustment and clearly demonstrates that we performed a simple hyperparameter search.

---

> > ### Comment · Reviewer_hopm · 2024-11-26
> >
> > Thanks for answering my questions. My main concern was about comparing to baselines with similar data requirements  (such as AdaMerging) which has been addressed, so I will raise my score accordingly.

---

> > > ### Author Response · Authors · 2024-11-27
> > > **Reply to reviewer hopm**
> > >
> > > Thank you for taking the time to review our work and providing constructive feedback. If you have any further questions or comments, please do not hesitate to let us know.

---

### Comment · Area_Chair_R1CM · 2024-11-22
**Discussion**

Dear reviewers,

The authors have responded to your reviews.

Until November 26th @ 2359 (AOE time) reviewers and authors can freely exchange responses, so if there any clarifications you require from the authors, now is the time to seek them!

Best,

AC

---

### Meta-Review · Area_Chair_R1CM · 2024-12-17

**Metareview:**

This paper is concerned with task arithmetic, and to this end the authors propose the $\tau$-Jacobian product ($\tau$Jp) metric which relates to weight disentanglement. They show that by adding a regulariser to minimise this quantity they can improve the performance of task arithmethic in practice.

In terms of strengths, reviewers thought this work was well-motivated,  and the $\tau$Jp metric was seen as "novel", "well-motivated", "reasonable", and "interesting". The proposed regularisation method was well-received for its practical benefits (it eliminates the need for tuning hyperparameters at inference time), and its simplicity. The experiments and results were lauded.

For weaknesses, there were concerns about a lack of novelty in comparison to [1], a lack of theory behind the success of the proposed regulariser, computational cost, and a lack of experiments in terms of baselines and non-image domains.

The authors did well in responding to reviews, and provided additional experiments. As 3/4 of the reviews were borderline (8,6,5,5) post-rebuttal I encouraged the reviewers to have a discussion to move towards a more decisive opinion. While none of the borderlines explicitly declared either way, the resulting discussion was useful for my decision-making. Two reviewers echoed their appreciation of this paper in terms of the experiments and practical benefits, with another reiterating a lack of theoretical novelty.

I think the strengths of this paper outweigh the weaknesses, and it provides significant added value from [1]. My recommendation is for  acceptance (poster).

[1] Guillermo Ortiz-Jimenez, Alessandro Favero, and Pascal Frossard. Task arithmetic in the tangent space: Improved editing of pre-trained models. Advances in Neural Information Processing Systems, 2023.

**Additional Comments On Reviewer Discussion:**

Reviewer hopm believed that certain experiments and discussions were missing from the paper. These were provided by the authors and they raised their score from 6->8. Reviewer onTx had a good back-and-forth with the authors, requesting additional experiments and results of computational cost. The authors did well at responding to these, and the reviewer raised their score from 3->5 which to me was a significant achievement given their initial stance. Reviewer rGLy had concerns (including a comparison of regularisers) which were addressed by the authors, raising their score from 5->6. After an exchange, Reviewer n6ZH remained unconvinced that this work was a significant advancement of [1] so kept their score as a 5. I note that the last two reviewers mentioned had low confidence scores (2). I think the authors did a great job at convincing three reviewers to up their scores. Most concerns were addressed apart from issues of novelty relating to [1] but as the experiments and method presented were so well-appreciated I believe this is a significant enough improvement to warrant acceptance.

---

### Decision · Program_Chairs · 2025-01-22

Accept (Poster)